# An *in vitro* method for inducing titan cells reveals novel features of yeast-to-titan switching in the human fungal pathogen *Cryptococcus gattii*

**Lamin Saidykhan**[1,2], **Joao Correia**[1], **Andrey Romanyuk**[1,3], **Anna F. A. Peacock**[3], **Guillaume E. Desanti**[1], **Leanne Taylor-Smith**[1], **Maria Makarova**[1], **Elizabeth R. Ballou**[1,4]*, **Robin C. May** [1]*

**1** Institute of Microbiology & Infection and School of Biosciences, University of Birmingham, Edgbaston, United Kingdom, **2** Division of Physical and Natural Science, University of The Gambia, Brikama, The Gambia, **3** School of Chemistry, University of Birmingham, Edgbaston, United Kingdom, **4** MRC Centre for Medical Mycology, University of Exeter, Exeter, United Kingdom

* e.ballou@exeter.ac.uk (ERB); r.c.may@bham.ac.uk (RCM)

**Data Availability Statement:** All relevant data are within the manuscript and supporting files.

## Abstract

Cryptococcosis is a potentially lethal fungal infection of humans caused by organisms within the *Cryptococcus neoformans/gattii* species complex. Whilst *C. neoformans* is a relatively common pathogen of immunocompromised individuals, *C. gattii* is capable of acting as a primary pathogen of immunocompetent individuals. Within the host, both species undergo morphogenesis to form titan cells: exceptionally large cells that are critical for disease establishment. To date, the induction, defining attributes, and underlying mechanism of titanisation have been mainly characterized in *C. neoformans*. Here, we report the serendipitous discovery of a simple and robust protocol for *in vitro* induction of titan cells in *C. gattii*. Using this *in vitro* approach, we reveal a remarkably high capacity for titanisation within *C. gattii*, especially in strains associated with the Pacific Northwest Outbreak, and characterise strain-specific differences within the clade. In particular, this approach demonstrates for the first time that cell size changes, DNA amplification, and budding are not always synchronous during titanisation. Interestingly, however, exhibition of these cell cycle phenotypes was correlated with genes associated with cell cycle progression including *CDC11*, *CLN1*, *BUB2*, and *MCM6*. Finally, our findings reveal exogenous p-Aminobenzoic acid to be a key inducer of titanisation in this organism. Consequently, this approach offers significant opportunities for future exploration of the underlying mechanism of titanisation in this genus.

## Author summary

*Cryptococcus gattii* is a fungal pathogen that causes lethal infections in humans and other animals. Upon entry to the lung, some (but not all) cryptococcal cells are induced to become so-called 'titan cells'; huge cells with thickened cell walls, altered capsule and highly duplicated DNA. As a key virulence determinant, titan cells can manipulate the

**Funding:** This work was supported by the UK Biotechnology and Biological Research Council through awards BB/R008485/1 to RCM and BB/M014525/1 to ERB. ERB is supported by a Sir Henry Dale Fellowship jointly funded by the Wellcome Trust and the Royal Society (211241/Z/18/Z). LS is supported by a PhD scholarship from the Islamic Development Bank. The funders had no role in study design, data collection and analysis, decision to publish or preparation of the manuscript.

**Competing interests:** The authors have declared that no competing interests exist.

host immune system (avoiding phagocytosis and triggering a non-protective immune response) and enhance dissemination to the brain. Thus far, the process of forming titan cells has been largely studied in the related species *C. neoformans*. Here we describe a new *in vitro* method to induce titan cells in *C. gattii*. Using this approach, we discovered the novel, asynchronous progression of cell cycle phenotypes during titanisation and showed how titan formation is strongly influenced by population density as well as the external environment; in particular, the presence of the molecule p-Aminobenzoic acid.

## Introduction

*C. neoformans* and *C. gattii* are two pathogenic species of *Cryptococcus* that cause invasive cryptococcosis in immunocompromised patients as well as immunocompetent individuals [1–3]. Upon inhalation into the lungs, *Cryptococcus* is exposed to a repertoire of hostile host factors (e.g., elevated temperature, nutrient deprivation, higher physiological $CO_2$ and hypoxia), [4,5] which trigger adaptive phenotypes such as the formation of titan cells. Titan cell formation is a dramatic morphological change as cryptococcal haploid yeast cells (5–7μm) transform into enormous polyploid titan cells (50–100μm in diameter) [6,7]. This atypical morphotype is characterized by many attributes such as enlarged cell size, thicker cell wall, and altered capsule composition, which confer resistance to host immune defence and enhance survival in the host [8–10].

*C. neoformans* titan cells have been clinically observed [11,12], studied *in vivo* [6,7] and recently induced *in vitro* [13–15]. The discovery of *in vitro* induction protocols is considered a major breakthrough, as efforts employed towards understanding the biology and mechanism underlying titanisation were impeded by the ethical and technical challenges associated with using animal models. With the invention of in *vitro* induction models, key regulators of titan cell formation have been identified [16]. Exogenous factors including host-specific environmental conditions [such as physiological temperature (37˚C) and $CO_2$ level (5%) coupled with hypoxia, nutrient limitation etc.] and other cues (including serum) have been identified as essential signals, while low-density growth (via quorum sensing) serves as a regulator of titan cell formation [14,15,13]. Although many molecular details remain unclear, endogenous genetic regulators and transcription factors associated with titan cell formation include several members of the cAMP/PKA pathway (including Gpa1, Cac1, Ric8, Pka1 Rim101 and Usv101) [15,13,17]. Titan cell formation is also dependent on altered cell cycle progression, as cells undergo endoreduplication (replication of nuclear genome in the absence of mitosis) to form dramtically enlarged, polyploid cells [16]. During titanisation, *C. neoformans* yeast cells initially exhibit a stationary-phase growth during which DNA duplication occurs to form unbudded 2C/G2 arrested cells [18]. Thereafter, the cell cycle regulator Cln1 is suppressed, allowing DNA re-replication, creating polyploid titan cells. After titan cells have been formed, Cln1 is then upregulated in order to release cells from G2 arrest and enable them to reenter the cell cycle and produce daughter cells.

Although titanisation is thought to be a major virulence factor of *C. gattii* [19–21], the defining attributes and underlying mechanism of titanisation have thus far been mainly characterized in *C. neoformans* [22–24]. Here we describe a facile *in vitro* induction approach that reveals a novel strategy for titanisation in *C. gattii*. Specifically, this method reveals that the *C. gattii*/VGIIa strain R265 deviates from the usual synchronous occurrence of cell enlargement and polyploidization that occurs in *C. neoformans*. Instead, R265 delays DNA endoreduplication, growing to around 30μm as a haploid cell over the first 3 days of induction before

endoreduplicating its DNA to become a uninucleate polyploid cell. Secondly, unlike *C. neoformans*, R265 titan cells produce daughter cells before reaching their critical cell size, after which they permanently cease budding unless they are exposed to conditions compatible with normal yeast growth. Although occurrence of these cell cycle phenotypes is asynchronous, these behaviors during titanisation are correlated with the expression levels of genes associated with cell cycle progression such as *CLN1*, *CDC11*, *MCM6* and *BUB2*. By studying offspring from crosses between R265 and other *C. gattii* strains, we show that the propensity to titanise is most likely a highly polygenic trait and identify a number of hybrid strains in which *in vitro* titanisation levels approach 100%, opening the door to future molecular studies of this biological phenomenon.

Finally, we discovered that the widespread metabolite and UV-scavenger p-Aminobenzoic acid (pABA) is required for maximal titanisation *in vitro*, providing a molecular inroad to future investigations of the titanisation pathway.

## Results

### Titan cells are induced by growth in RPMI medium

While characterizing co-culture of *C. gattii* with an alveolar macrophage cell line (MH-S) at a low multiplicity of infection (MOI [5:1]), we noticed that exposure of low density R265 cultures to sterile RPMI growth media at 37°C in an atmosphere of 5% $CO_2$ induced dramatic cell size increase within 24 hrs (Fig 1A and 1B). Although the phenotype was more prominent in the presence of serum supplemented-RPMI (median cell body size: 13.57μm vs 9.8μm, S1A Fig), the ability of serum-free RPMI to produce these enlarged cells provided the opportunity to probe the molecular basis of titanisation in this species using a simple, chemically-defined medium. Extending the incubation period to seven days enabled these cells to achieve cell

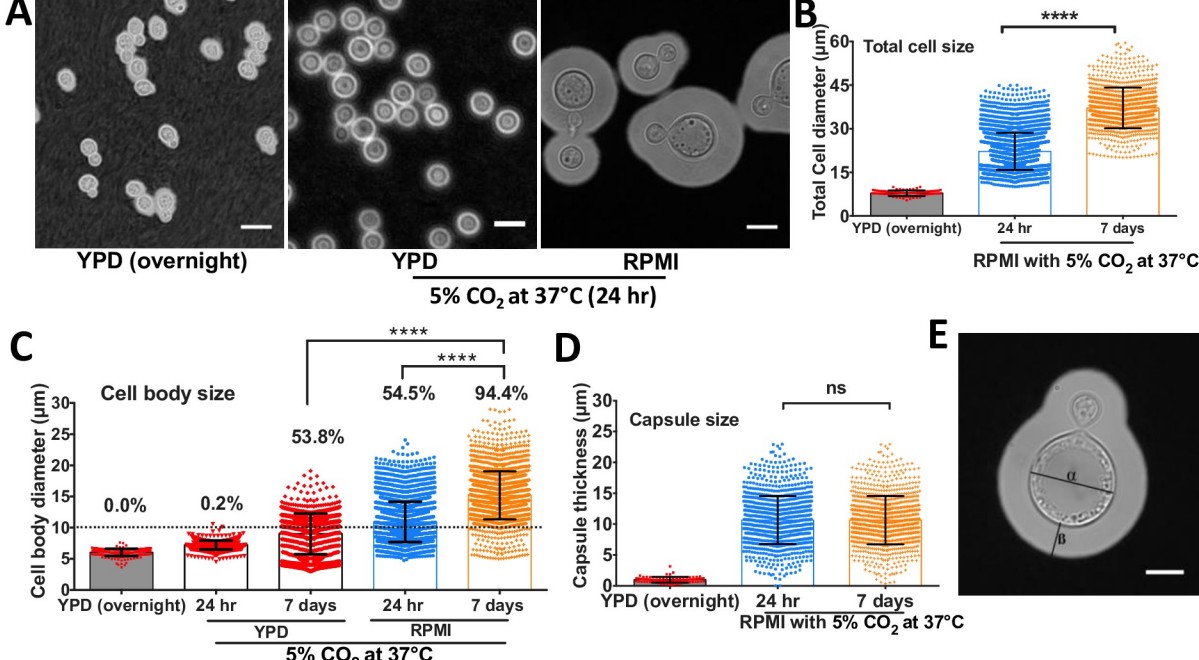

**Fig 1. *C. gattii* (R265) exhibits cell body and capsule enlargement in response to growth in RPMI.** A) Micrographs of R265 cells grown in YPD overnight at 25°C in atmospheric conditions (left panel), or at 37°C in an atmosphere of 5% $CO_2$ either in YPD (middle panel) or RPMI (right panel). Scale bar = 15μm. B) Total cell diameter (capsule included), C) cell body diameter (percentages represent percentage of cells more than 10μm in diameter) and D) capsule thickness of R265 cells all significantly increase after growth in RPMI at 37°C in 5% $CO_2$. The graphs represent at least 3 biological experimental repeats, and Kruskal-Wallis test was used to determine significance where **** = p<0.0001 and ns = p>0.05. E) Cellular morphology of R265 enlarged cells: cell body diameter (α) and capsule thickness (ß). Scale bar = 15μm.

bodies of up to 30μm in diameter even in the absence of serum, resulting in a population with a median size significantly higher than yeast cells grown in YPD at 37°C in an atmosphere of 5% $CO_2$ for 7 days [median size: 13.7μm (5.1–29.8) vs 10.17μm (3.00–19.11) (p<0.0001)] (Fig 1C and 1E [α = cell body diameter], S1B Fig). Notably, the pH of both media remained close to neutral throughout the assay, ruling out a secondary impact on titan cell formation as a result of pH stress (S1C Fig).

The cryptococcal yeast-titan transition occurs concomitantly with increasing capsule thickness [19]. Therefore, we characterized the capsule size (Fig 1D and 1E [ß = capsule size]) of our *in vitro*-generated R265 enlarged cells. Relative to the YPD-grown capsule thickness, R265 enlarged cells demonstrated significantly thicker capsule [median capsule thickness: 10.28μm (0.34–22.9) vs 1.00μm (0.2–3.1), p<0.0001]. Although the cell body size progressively increased with induction time from 24 hr to 7 days (Fig 1C), the capsule size of the enlarged cells reached a plateau at 24 hours and remained at a median size of 10.33μm (1.0–22.9) for the remaining six days of the assay (Fig 1D). This suggests that the enlarged cells achieve their maximum capsule size much earlier than their maximum cell body size.

Titan cells exhibit altered cell wall composition [13,15,25,26]. Consequently, we characterized the cell wall of R265 *in vitro*-derived enlarged cells by staining for chitin with calcofluor white (CFW) [15]. In line with previous studies, we observed a significant increase (p<0.0001) in the chitin content of enlarged induced cells via flow cytometry that was also evident via fluorescence microscopy (CFW, Fig 2A, 2B and 2C). In addition to displaying a titan-like capsule, cell body and cell wall, *C. gattii* (R265) enlarged cells bear a single large vacuole occupying

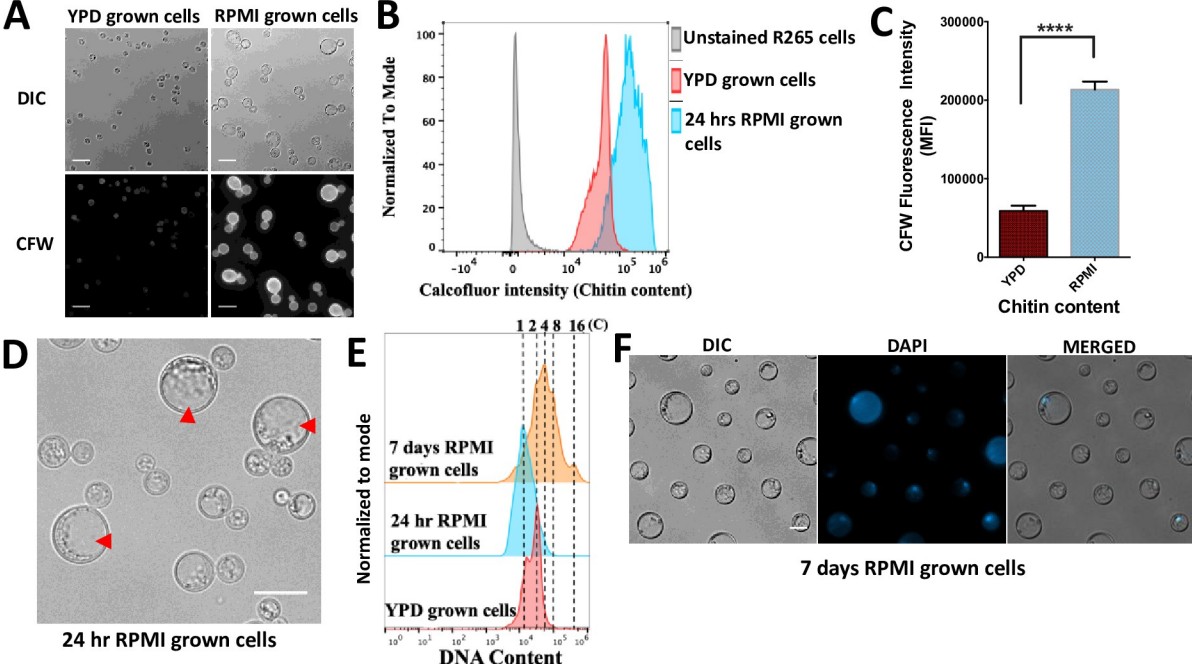

**Fig 2. The cell wall chitin content and ploidy of enlarged cells are typical of titan cells.** *In vitro* RPMI-generated enlarged cells displayed a significantly higher chitin level relative to YPD grown R265 cells, as indicated by the calcofluor white (CFW) fluorescence intensity shown using microscopy imaging in (A), via flow cytometry in (B) and graphically in (C, Median Fluorescence Intensity, MFI) (p<0.0001). Scale bar = 15μm. Statistical significance was confirmed by Two-tailed *t*-test. D) Micrograph showing single large central vacuole of *in vitro*-generated enlarged cells. E) Ploidy measurement of RPMI-generated enlarged cells based on flow cytometry analysis of DAPI staining. YPD grown cells (red) gated as 1C, 2C (haploid) were used as a control to determine ploidy (DNA content) of enlarged cells after 24 hr (blue) and 7 days (orange) of induction. F) Micrographs showing the uninucleate nature of R265 titan cells upon staining with DAPI to visualize the nucleus. Scale bar = 10μm.

almost the entire cytoplasmic space (Fig 2D), a characteristic previously described in both *in vivo* and *in vitro*-derived *C. neoformans* titan cells [7,13].

In addition to morphological changes, the yeast-titan transition in *C. neoformans* involves a switch from a haploid to highly polyploid state [6,7,13,15]. Thus, we evaluated the ploidy of RPMI *in vitro*-generated *C. gattii* enlarged cells. Whilst YPD grown yeast cells typically displayed a mix of cells with 1C or 2C DNA content (depending on which phase of the cell cycle they are in), RPMI-induced cells displayed DNA content ranging up to 16C after 7 days of induction (Fig 2E). By DAPI staining the nucleus and visualizing by microscopy, we confirmed the enlarged cells were uninucleate (Fig 2F). Thus, R265 titan-induced cells exhibit all the key features of *bona fide* titan cells: cell enlargement, a large vacuole, altered cell wall composition and high ploidy. Interestingly, however, we noted that in the case of R265, cell enlargement and increased DNA content do not necessarily occur at the same time. Indeed, R265 cells achieve significant cell body enlargement within 24hrs, but DNA content does not exceed 2C until much later (Fig 2E). Taken together, the structural attributes exhibited by the *in vitro* RPMI-induced titan cells are typical of true titan cells.

## Titan cell formation is inversely correlated to cell density

Previous work in *C. neoformans* has shown that titan cells are preferentially produced at low cell density [7,14,15,27]. We tested whether the same was true of R265 by growing yeast cells in RPMI at five decreasing inoculum concentrations ($10^6$, $5 \times 10^5$, $5 \times 10^4$, $5 \times 10^3$ and $10^3$ cells/mL) and incubating for 24hr at 37°C in 5% $CO_2$. The formation of titan cells was maximal [87.9% titan cells (>10μm)] at an initial inoculum of $5 \times 10^3$ cells/mL, producing cells with a median cell body size of 14.8μm (4.2–26.28) (Fig 3A). The percentage of titan cells decreased with increasing cell densities, indicating that in *C. gattii*, as in *C. neoformans*, titan induction occurs primarily in low density cultures.

## Quorum sensing effect

The density dependence of titanisation is suggestive of a quorum sensing effect. Consequently, we tested the impact of the putative cryptococcal quorum sensing molecule Qsp1, a short peptide previously reported to inhibit titanisation in *C. neoformans* cells *in vitro* [14,15]. By adding the Qsp1 peptide (NFGAPGGAYPW) at 20μM to RPMI, both the median cell body size of the R265 cells and the proportion of titan cells was significantly reduced [median cell size: 13.35μm vs 11.74μm (p<0.0001); percentage titan cells: 88.3% vs 75.8%]. In contrast, a 'sequence-scrambled' Qsp1 peptide (AYAPWFGNPG) had no effect (Fig 3B). Interstingly, the active peptide also impacted capsule growth by reducing capsule size of the titan induced cells significantly (see micrograph images of Fig 3B). Taken together, this indicates that titan cell formation in *C. gattii* is modulated by quorum sensing via the Qsp1 pathway, as previously described in *C. neoformans* [14,15].

## High density growth in RPMI produces large, haploid, non-titan cells

The impact of culture density led us to ask what happens to R265 in non-titanising (high density) conditions. To answer this, we inoculated R265 cells in RPMI at $10^6$ cells/mL and measured cell body size, capsule size, and ploidy over 7 days. At high density, there was a statistically significant increase in cell body size from 24 hr to 7days (median size: 9.0μm vs 10.5μm, p<0.0001) (Fig 3C), but cells did not become the very large titans that appear in low density cultures (Fig 1C). At both time points in these high-density conditions, cell enlargement typically did not surpass 15μm, with 0.1% (4/4302) and 1.3% (46/3489) of cells scoring >15μm at 24 hr and 7 days respectively (Fig 3C). The 7-day cultures were fully growth-

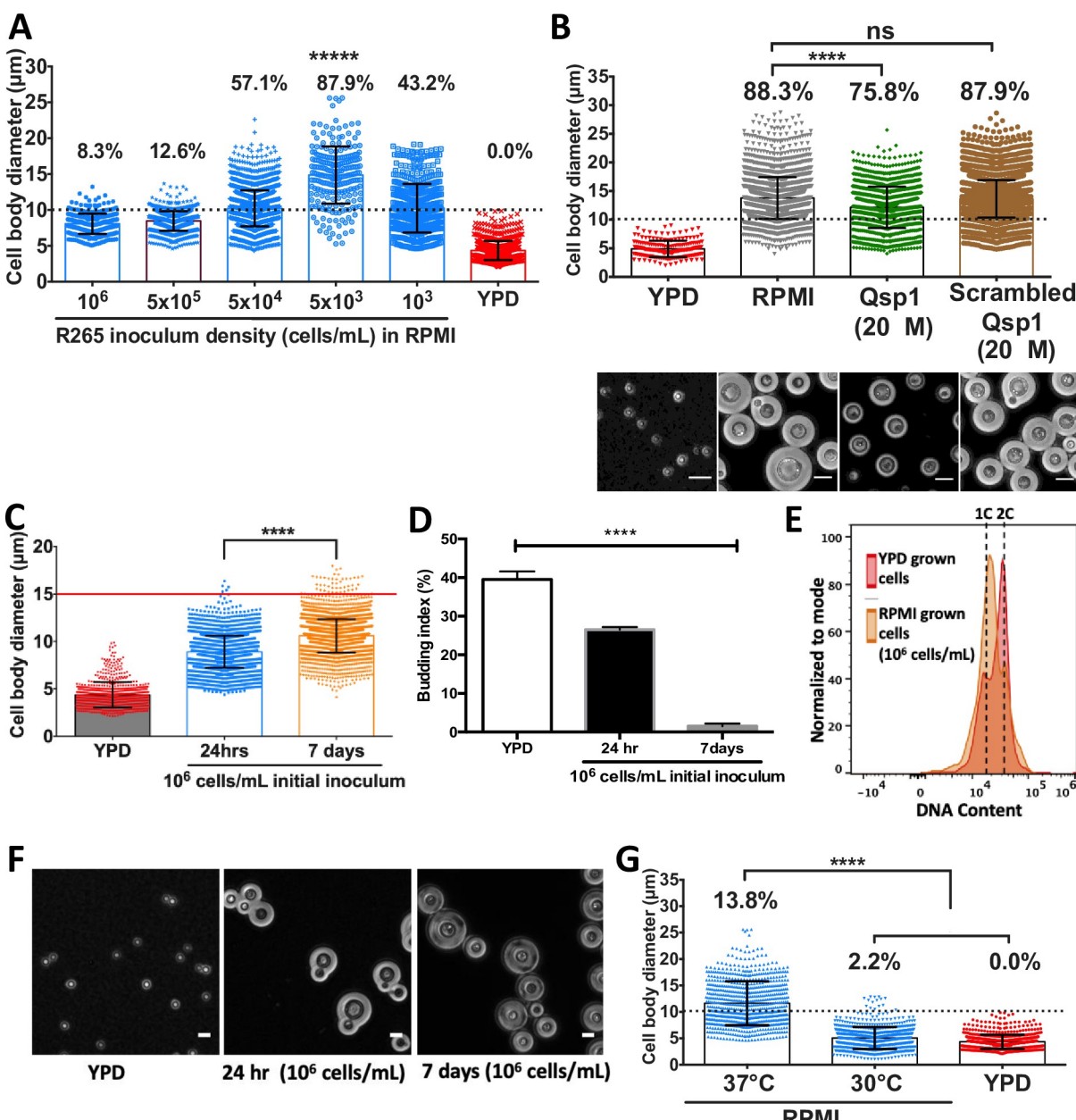

**Fig 3. Effect of cell density and environmental conditions on R265 titan cell formation.** A) The effect of cell density on cell enlargement was determined by growing R265 cells in titan-inducing conditions at varying concentrations (between $10^6$ and $10^3$ cells/mL, as indicated) and cell size being measured 72 hrs later (percentages indicate proportion of the population >10 μm). B) The role of the quorum sensing peptide, Qsp1 (on titan cell formation) was determined by growing R265 yeast cells in RPMI supplemented with the intact and scrambled peptide (control) at 37˚C with 5% $CO_2$ and measuring the cell body size after 72 hr incubation. Microscopy images illustrating the morphological differences in titan cell production between these three conditions (RPMI, intact Qsp1 and scrambled Qsp1) are also shown. Maximum cell enlargement capacity (C), budding index (D) and ploidy (E) of R265 cells in RPMI at $10^6$ cells/mL (high density growth) at 37˚C with 5% $CO_2$ for 24 hr and 7 days. Budding index was expressed as percentage of budded cells per total number of cells. At least >3000 cells were analysed for each sample from two independent repeats and significance was determined by one-way ANOVA (**** = p<0.0001). (E) Cells recovered from 7 day high-density induction cultures were analysed for ploidy (orange) which was consistent with a 1C and 2C haploid DNA content as found for the YPD grown cells (red). F) Microscopy images depicting the budding index of cells grown in YPD (left panel), RPMI for 24hr (middle panel) and RPMI for 7days (right panel). Scale bar: 10μm. G) Impact of temperature and 5% $CO_2$ growth: cell body diameter of R265 cells after 24hrs of growth in RPMI with 5% $CO_2$ at 37˚C or 30˚C. The graphs represent at least 3 biological experimental repeats, and Kruskal-Wallis test was used to determine significance where **** = p<0.0001.

arrested, as evidenced by their very low budding index (the number of mother cells producing buds) of 2.1% (67/3122) (Fig 3D and 3F). These 7-day high-density cells did produce dramatic capsule (Fig 3F), consistent with the known effects of RPMI media and $CO_2$, which have been employed for capsule induction in *Cryptococcus* [19,27–30]. Interestingly, unlike cells grown in low-density titan-inducing conditions, cells grown at high density remained haploid over the entire 7 day period (Fig 3E). We conclude that high-density long-term culture in RPMI induces a degree of cell enlargement in R265, but these large cells are distinct from the true titan cells that appear in low density culture.

### Impact of temperature and $CO_2$ on titan cell formation

*Cryptococcus* responds to human physiological temperature (37˚C) by exhibiting a variety of morphological changes including capsule elaboration, cell body enlargement and cell shape alteration [13,30,31]. Hence, we investigated the role of temperature on R265 yeast-titan cell transformation by comparing cells grown in the presence of 5% $CO_2$ at either 37˚C or 30˚C. At 30˚C incubation, no titan cells were generated and the median cell body size [5.03μm (1.04–12.85)] was significantly lower than at 37˚C [11.17μm (4.04–28.6)]. A similar trend was observed for the proportion of titan cells (13.8% vs 2.2%) (Fig 3G). Thus, elevated temperature was essential to produce titan cells in our *in vitro* protocol.

The most dramatic biological response of *Cryptococcus* to $CO_2$ is capsule biosynthesis which occurs concurrently with cell body enlargement [19,29]. Consequently, we compared cells grown at 37˚C in 5% $CO_2$ with those grown under normal atmospheric conditions at 37˚C. Relative to growth in 5% $CO_2$, the proliferation of R265 was significantly inhibited in ambient atmosphere [Mean CFU: $CO_2$ = 28.5 x$10^5$ cells/mL vs $CO_2$- free 0.29 x$10^5$ cells/mL; (p<0.001)] and no titan cells were observed in this condition (S2A Fig). Thus, both elevated temperature and high $CO_2$ are required for *in vitro* titan cell formation.

### In R265, cell enlargement is asynchronous with ploidy

To characterise the kinetics of cell size and ploidy changes more fully, we carried out a detailed time course analysis of R265 cells over a period of three weeks. By day 3 of induction, R265 cells showed significantly enlarged cell body diameter as compared with non-induced (YPD grown) cells [median size: 13.7μm (5.1–29.8) vs 6.5μm (4.9–8.89); (p<0.0001)] with 81.5% (5241/6431) of cells larger than 10μm (Fig 4A). Despite this size increase, for the first three days all cells were 1C or 2C (reflecting a haploid cell cycle) (Fig 4B). The population reached the maximum cell size on day 5, and from day 5 to day 7, there was no change in the size of induced cells [median size: 14.9μm at day 5 and 14.7μm at day 7 (p = 0.7656)]. However, the ploidy of these cells increased, with tetraploid (4C) cells apparent at day 5 and cells exhibiting DNA content of up to 16C present by day 7. In parallel, although the maximum size of induced cells no longer increased at this late stage of incubation, the proportion of the population with a large cell phenotype rose from 92.8% to 99.3% (p<0.0001) (Fig 4A). Thus, it appears that: a) size increase and ploidy increase are separable phenotypes during titanisation of R265, b) true titan cells (large and polyploid) appear only after around 5 days of induction and c) their maximum cell size is achieved rapidly during titanisation, but the proportion of cells adopting this fate rises steadily over a long time period.

### The polyploid titan cells are unbudded

Within our *in vitro* model, after 3 days of induction, the R265 titan cells completely stop budding despite being polyploid (Fig 4C and 4D). This suggests that budding and DNA increase are decoupled, consistent with endoreduplication. During the first 3 days, budding index fell

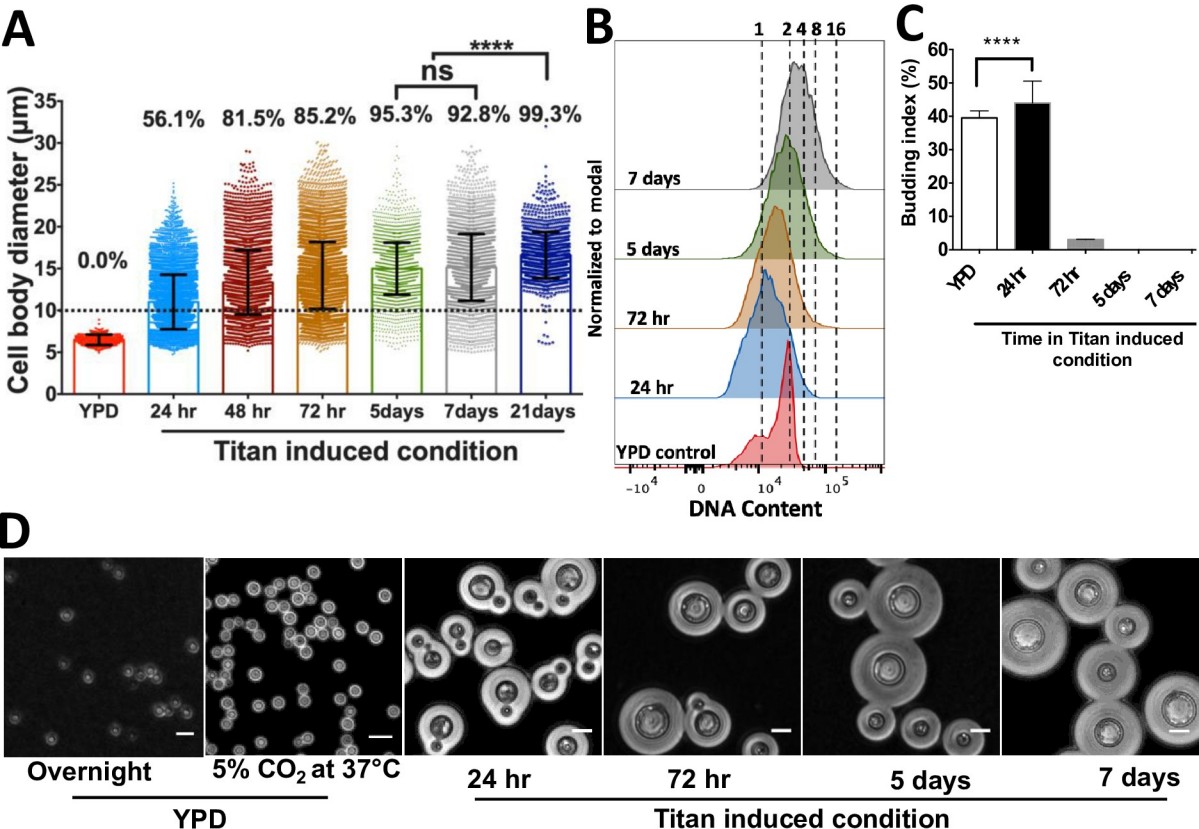

**Fig 4. Cell enlargement, polyploidization and budding occur at different periods during titan induction in R265.** A) Cell body diameter of R265 titan cells over prolonged induction (24 hr to 21 days) as compared to YPD grown cells. Enlarged cells were recovered at different time points, fixed and measured. The error bars represent median of 3 biological repeats, where **** = p<0.0001. B) Samples from all the induction time points were fixed, DAPI stained and analysed for ploidy by flow cytometry with reference to YPD grown cells, which were used to gate for 1C and 2C DNA content. C) Budding index of cells recovered from YPD and titan inducing condition. Budding index was expressed as percentage of budded cells per total number of cells. At least 3000 cells were analysed for each sample (the graph represents three biological repeats and significance was confirmed by one-way ANOVA, where **** = p<0.0001 D) Microscopy images showing the budding nature of cells obtained from YPD or at various timepoints after titan induction. Scale bar: 15μm.

from 44.4% (2552/5749 cells) at 24 hr to 2.8% (26/933) by day 3 and 0% (0/2971) by the fifth day of induction (Fig 4C). Hence, we termed the period before 3 days of induction as the budded phase and the later period as the unbudded phase. During the budded phase, cells predominantly exhibited a 1C DNA content (Fig 4B). However, during the unbudded phase (3 to 7 days) cells rapidly became polyploid, with DNA content rising from 2C to 4C to 16C (Fig 4B). The 1C DNA content during the budded phase, coupled with cell enlargement, suggests that the cells spend longer in the G1 phase of their cell cycle during titan induction.

## Unbudded titan cells are viable and metabolically active

The unbudded state of *C. gattii* (R265) titan cells prompted us to investigate whether this cell cycle arrest occurs as a consequence of nutrient deprivation. By filtering 7 day old titan-induced cultures, we obtained the largest (>20μm) titan cells and re-cultured them on a rotary wheel at 20 rpm for i) 2 hr or ii) overnight at 25°C in YPD broth. These cells remained unbudded during the first 2 hrs (the generation period of *Cryptococcus* yeast) (Fig 5A) but after overnight incubation produced daughter cells (Fig 5B and 5C) resembling YPD-grown yeast cells in size and morphology (round-shaped with small capsule size) (Fig 5D). Time-

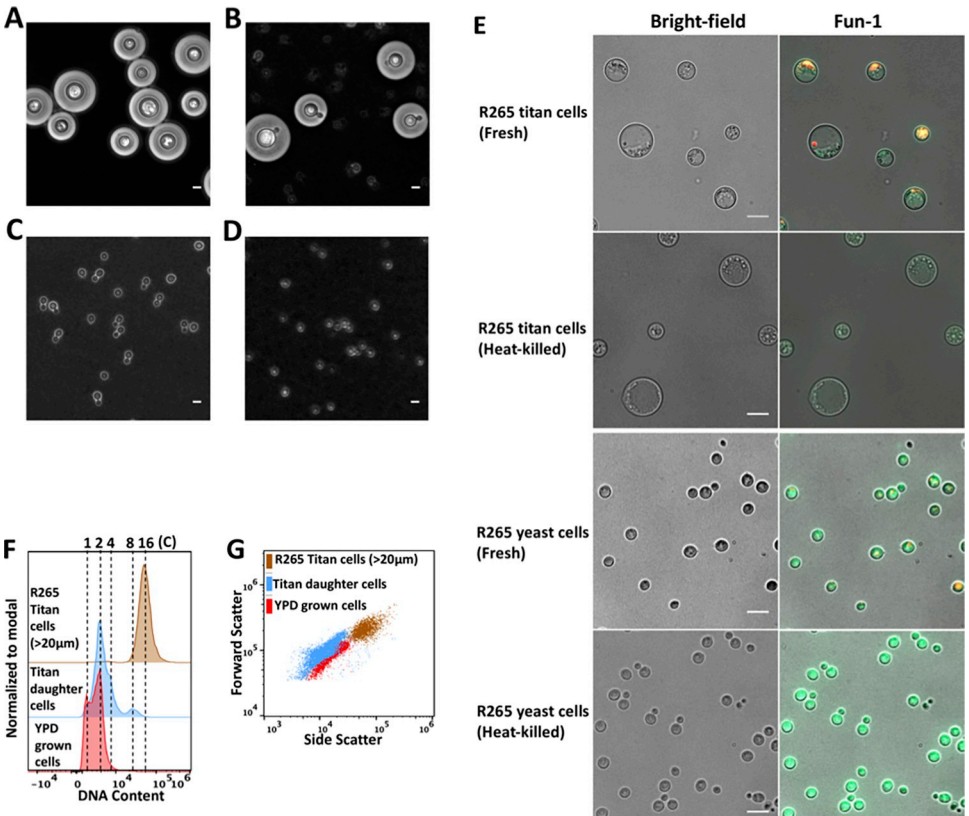

**Fig 5. Characterization of daughter cells and metabolic state of R265 titan cells.** 7 day old R265 titan cells were purified using a 20μm cell strainer (A) and then B) re-cultured overnight in YPD to induce budding. C) After 24hrs, daughter cells of R265 titan cells were isolated by filtration of the titan culture (through a >15μm cell strainer) and microscopically compared with D) YPD grown yeast R265 cells (control). E) Microscopy images demonstrating the metabolic activity titan cells from 7 days old cultured (fresh vs heat-killed) and YPD grown yeast cells (fresh vs heat-killed). The cells were stained with the metabolic reporter dye Fun-1 which is converted from yellow-green to orange-red by metabolically active cells. Scale bar = 10μm. F) DNA content of titan-derived daughter cells (Blue) cells as compared to YPD grown (red) and >20μm R265 titan cells (brown). G) Size distribution of daughter cells (blue), YPD grown (red) and >20μm R265 titan cells (brown).

lapse microscopy revealed that recovered titan cells start budding after 2 hours (S1 Movie). The active proliferation in YPD (nutrient rich media) of the unbudded titan cells suggest that their exhibition of growth arrest is due to nutrient depletion after the long incubation. In line with this, proliferation could also be observed when these unbudded titan cells were re-cultured in fresh RPMI (S2B Fig).

We tested whether unbudded R265 titan cells are quiescent, but metabolically active, by using the dye Fun-1, whose emission wavelength is converted from yellow-green to orange-red colour by metabolically active cells. Unbudded titan cells were able to convert the FUN-1 dye in a manner that was comparable to actively-growing yeast cells (Fig 5E). Thus, both time-lapse imaging and metabolic profiling indicates that titan cells remain viable and metabolically active.

## Titan cells produce polyploid, yeast-sized, daughter cells

To investigate if the cellular similarities observed between titan-derived daughter cells and yeast cells extend to their ploidy, we characterized the DNA content of daughter cells relative

to their highly polyploid titan mother cells (>20μm) and haploid yeast cells. Despite having the cellular properties of yeast cells (Fig 5C and 5G), titan-derived daughter cells exhibited a higher DNA content than normal yeast, with most cells displaying either diploid (2C) or polyploid (>4C) DNA content (Figs 5F and S3A shows gating strategy). Taken together, the delayed DNA replication of R265 titan cells (Fig 4B) coupled with the production of polyploid daughter cells (up to 8C) (Fig 4E) is indicative of high ploidy elasticity in *C. gattii* titan cells.

By re-culturing titan-derived daughter cells in titan inducing condition, we assessed the ability of these 'second generation' cells to return to a titan state. Unlike the 'original' titan cells, these titan-derived daughters increased their DNA content within 24 hours, achieving genome sizes of 16C and 32C at 24hr and 7 days respectively (S3E Fig). As with the original mother titan cells, titan-induced daughter cells underwent budding during the early induction period but formed unbudded titan cells by day 7 of induction (S3B and S3C Fig).

## The titan cell cycle phenotypes are correlated with the expression of genes involved in cell cycle progression

The manner in which R265 progressively exhibits cell-cycle-associated phenotypes (cell enlargement, budding, DNA replication and finally growth arrest) to form unbudded polyploid titan cells led us to question the underlying cell cycle timing. We extracted RNA from R265 titan induced cells for 24 hr, 3 days, 5 days and 7 days of our *in* vitro protocol and then investigated the expression of a panel of cell cycle markers via quantitative RT-PCR (S1 Table).

In line with the phenotypic changes that we observe during titan cell formation, the cell cycle markers we examined showed a clear shift from budding and mitosis to DNA replication and eventually cell-cycle arrest (Fig 6). Thus, the *CDC11* gene, encoding a septin protein involved in bud formation, was highly expressed in YPD grown cells and for the first 24hrs of induction, but downregulated in the "unbudded phase" timepoints (Fig 6A). The expression of the G1 cyclin *CLN1* similarly was reduced at day 3 compared to 24 hrs, while expression was somewhat restored by day 5 (Fig 6B). In contrast, *MCM6* and *BUB2*, associated with DNA replication and G2 arrest respectively, both peaked at 7 days where the titan cells exhibit maximum polyploidy and remain unbudded (Fig 6C and 6D).

## P-Aminobenzoic acid is a major trigger of titanisation in RPMI

While comparing the titan induction capacity of RPMI [RPMI Medium 1640 (1X)] against Dulbecco's modified eagle medium (DMEM), we noticed that the full titan cell phenotype emerged in RPMI but not DMEM after 24 hrs of induction. Notably, cells enlarged in both media, but cells grown in DMEM never achieved the same size as those in RPMI and did not become polyploid. Since RPMI and DMEM media have a close chemical composition, we took advantage of this similarity and sequentially supplemented DMEM with RPMI-specific compounds with the aim of identifying an RPMI-specific factor that triggers titanisation in R265 cells. To more clearly distinguish between conditions, we used a higher threshold for titan cells in this assay (15 μm rather than 10 μm). After 24hrs of induction, 13.8% of cells in RPMI had a cell body size above 15 μm, whilst no cells of this size appeared in DMEM [median cell body size: 12.3 μm (4.1–28.8) vs 9.9 μm (4.5–14.9)] (Fig 7A). By 3 days of incubation, larger cells had appeared within the DMEM condition, but at a rate approximately three-fold lower than for RPMI (Fig 7B).

RPMI differs from DMEM in its amino acid composition and so we first supplemented DMEM with 'RPMI-specific' amino acids (L-Glutamic acid, L-Aspartic acid, L-Arginine, L-Glutathione, L-Asparagine and L-Proline). However, none of these conditions were sufficient to confer titan-inducing capacity to DMEM, even when supplemented at twice their

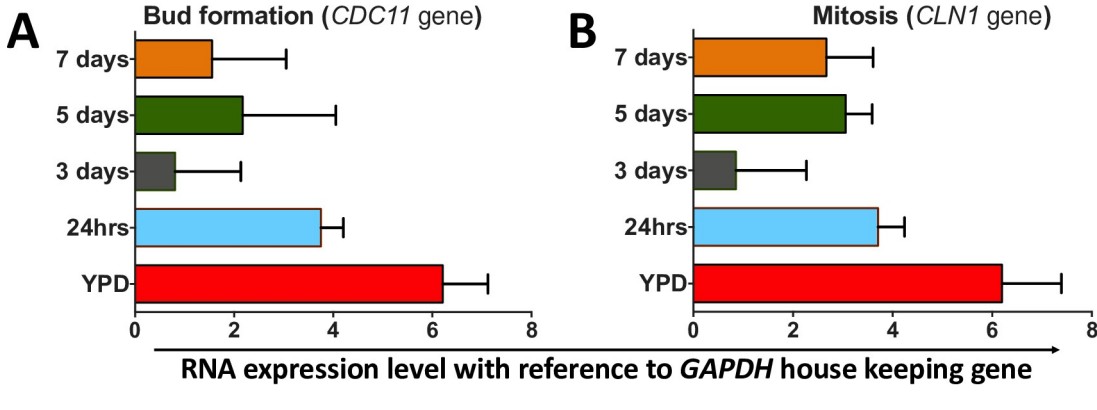

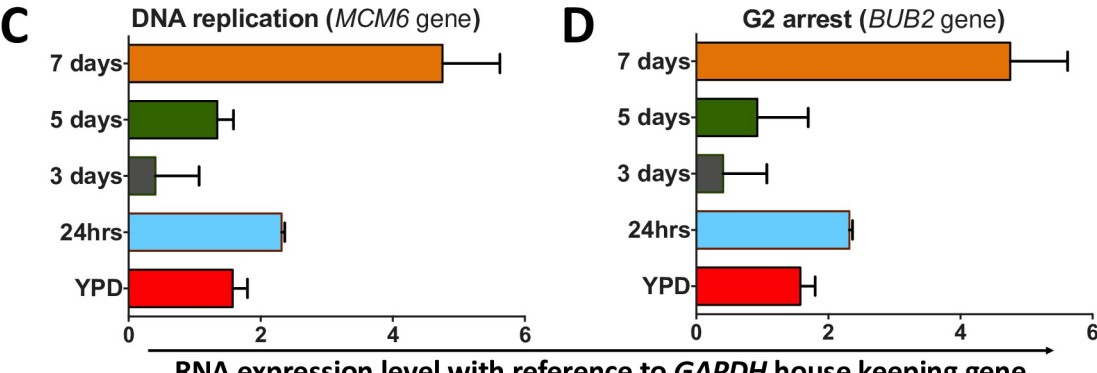

**Fig 6. Relative transcription of R265 cell cycle marker genes during titanisation.** Quantitative expression analysis of four different cell cycle associated genes in R265 grown in either YPD (control) or titanising conditions for the indicated time points. Expression is shown relative to the housekeeping gene GAPDH. Genes quantified were A) *CDC11 (CNBG_5339)*, involved in bud formation; B) *CLN1 (CNBG_4803)*, associated with balancing cell division and DNA replication [18]; C) *MCM6 (CNBG_5506)*, involved in DNA replication; and D) *BUB2 (CNBG_4446)*, involved in G2 arrest. The graphs represent 3 biological repeats (3 technical replicates each), with error bars depicting the standard deviation of *delta-delta* $C_T$ values.

normal concentration (Fig 7). We also increased the glucose concentration of DMEM from 1000mg/L to 2000 mg/L (RPMI concentration) but saw no significant impact on titan production (S1D Fig). Based on these results, we conclude that amino acid availability and glucose concentration are not the trigger for R265 titanisation.

We continued by testing three additional compounds that are present in RPMI but absent from DMEM: Vitamin B12, Biotin, and para-aminobenzoic acid (pABA). Whilst vitamin B12 or biotin supplementation into DMEM had relatively little effect, addition of pABA significantly increased the production of titan cells from 13.7% to 29.0% (median cell diameter = 11.94μm vs 12.82 μm, p <0.0001) (Fig 7B). Consequently, pABA appears to be a major trigger for titan cell formation in RPMI.

Finally, we noted that supplemental iron (in the form of Iron (III) nitrate nonahydrate-Fe $(NO_3)_3.9H_2O$) is present in DMEM but absent from RPMI. We therefore tested whether iron availability may actively inhibit titanisation. Indeed, adding iron to RPMI (Fe $(NO_3)_3.9H_2O$ +RPMI) slightly reduced the induction capacity of RPMI by ~4%. This was not surprising as iron limitation induces titan cell formation in *C. neoformans* [14]. In summary, therefore, the efficient induction of titan cells in RPMI likely results from the combined presence of pABA and the absence of supplementary iron.

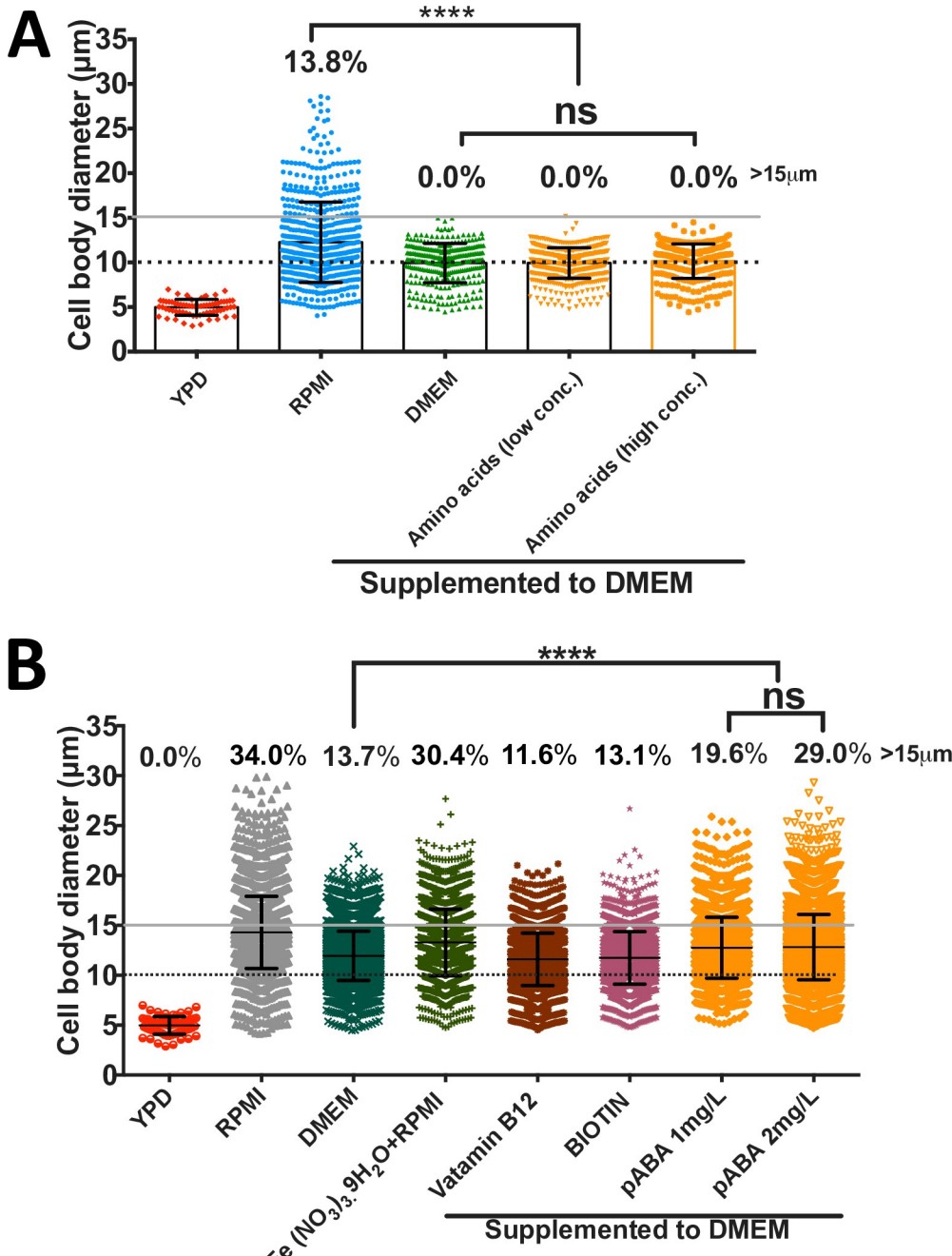

**Fig 7. The effect of RPMI-specific compounds on R265 titan cell formation.** A) RPMI-specific amino acids (L-Glutamic acid, L-Aspartic acid, L-Arginine, L-Glutathione, L-Asparagine and L-Proline) were added to DMEM either at the concentration used in RPMI ('low conc.') or two-fold higher ('high conc.') and tested for their capacity to trigger titan cell formation after 24 hr induction at 37°C in 5% $CO_2$. B) RPMI-specific compounds (Vitamin B12, Biotin, and para aminobenzoic acid (pABA)) were supplemented to DMEM at the levels present in RPMI and then evaluated for their capacity to induce titan cell formation after 3 days incubation at 37°C in 5% $CO_2$. The graphs are representation of 3 biological repeats and statistical significance was determined by Kruskal-Wallis test, where **** = $p < 0.0001$.

## Strain specificity

There is considerable evolutionary divergence between clades within the *Cryptococcus* genus and indeed the nomenclature of this group is rapidly changing in recognition of potential cryptic species [32]. To begin to assess variation in titanisation capacity, we screened 42 different cryptococcal isolates comprising 32 *C. gattii* species complex strains (VGI–VGIV), 8 *C. neoformans* strains (VNI and VNII), and 2 *C. deneoformans* strains (VNIV) for their capacity to form titan cells in our *in vitro* protocol (3 days incubation in RPMI with 5% $CO_2$ at 37˚C). Overall, the capacity to form titan cells in *C. gattii* strains was significantly higher than either *C. neoformans* or *C. deneoformans*. All 32 *C. gattii* strains produced titan cells, with an average of 60% titan cells at the end of the assay (Table 1). In contrast, only 4/8 (*C. neoformans*) and 0/2 (*C. deneoformans*) strains showed any level of titanisation (Table 1). Within the *C. gattii* species complex, VGII genotype strains (*C. deuterogattii*) displayed the highest titanisation capacity (averaging 80.0%) while VGIII (*C. gattii*) scored the lowest at 37.9% (Table 1).

**Table 1. Capacity to form titan cells across cryptococcal isolates.** Percentage of titan cells was determined based on capacity to enlarge >10μm (S4 Fig) and having >2C ploidy (S5 Fig).

| Species/strain | Genotype | Median size [range] (μm) | | % Titan cells |
|---|---|---|---|---|
| ***C. gattii*** | | | | |
| WM265 | VGI | 10.9 [4.1–28.4] | | 62.7 |
| WM179 | VGI | 11.3 [4.1–25.0] | | 67 |
| CBS8755 | VGI | 11.0 [4.6–17.7] | | 62.8 |
| C384 | VGI | 10.0 [4.2–27.9] | | 50.1 |
| NIH312 | VGI | 8.0 [4.1–24.3] | | 29.3 |
| B4546 | VGI | 8.9 [4.5–26.2] | | 35.3 |
| EJB11 | VGI | 10.8 [5.3–25.6] | | 65.5 |
| MMCO8-897 | VGI | 6.7 [4.4–16.3] | | 17.5 |
| CBS1508 | VGI | 9.0 [4.7–17.4] | | 27.3 |
| CBC1873 | VGI | 11.6 [5.8–22.7] | | 62.3 |
| | **Av. of VGI** | **9.82** | | **48.0** |
| R265 | VGIIa | 13.4 [5.1–29.6] | | 79.7 |
| CDDR271 | VGIIa | 12.9 [4.4–26.5] | | 84.7 |
| ENV152 | VGIIa | 10.6 [5.5–28.0] | | 54.8 |
| CDCF2866 | VGIIa | 11.3 [4.6–19.9] | | 89.1 |
| ICB180 | VGII | 12.7 [3.4–26.3] | | 82.6 |
| CBS10089 | VGII | 13.3 [5.6–25.1] | | 89.1 |
| CDCR272 | VGIIb | 13.5 [6.8–39.2] | | 90 |
| B7735 | VGIIb | 12.9 [5.5–26.3] | | 86.4 |
| EJB18 | VGIIc | 11.3 [4.7–19.5] | | 75.6 |
| EJB52 | VGIIc | 11.1 [4.8–19.5] | | 68.3 |
| | **Av. of VGII** | | **12.3** | **80.0** |
| CBS6955 | VGIII | 8.24 [4.34–17.9] | | 66.3 |
| CBS6993 | VGIII | 9.9 [4.8–22.6] | | 45.7 |
| CBS1622 | VGIII | 5.4 [3.3–10.9] | | 4.4 |
| WM1243 | VGIII | 10.1 [4.6–22.4] | | 50.7 |
| B13C | VGIII | 10.0 [4.4–23.5] | | 50 |
| CA2350 | VGIII | 8.8 [4.7–21.7] | | 32.7 |
| CA1227 | VGIII | 7.1 [4.4–15.2] | | 15.4 |
| | **Av. of VGIII** | | **9.1** | **37.9** |

*(Continued)*

**Table 1.** (Continued)

| Species/strain | Genotype | Median size [range] (µm) | % Titan cells |
|---|---|---|---|
| WM779 | VGIV | 11.8 [4.2–25.5] | 66.5 |
| CBS1010 | VGIV | 12.6 [4.3–22.2] | 80 |
| B5742 | VGIV | 8.9 [4.6–21.8] | 37.3 |
| B5748 | VGIV | 10.7 [4.8–22.9] | 61.9 |
| CBS10101 | VGIV | 10.9 [4.4–19.6] | 62.7 |
| | **Av. of VGIV** | **12.2** | **61.7** |
| **All *C. gattii* (Average)** | | **10.75** | **60** |
| ***C. neoformans*** | | | |
| H99 | VNI | 7.7 [3.4–20.1] | 21.3 |
| Zc1 | VNI | 7.6 [4.46–15.9] | 5.8 |
| Zc8 | VNI | 12.4 [5.4–20.7] | 76.1 |
| Zc12 | VNI | 4.1 [2.1–7.0] | 0 |
| CBS8336 | VNI | 9.6 [4.6–24.8] | 39.9 |
| 125.91 | VNI | 7.2 [4.6–16.3] | 7.9 |
| | **Av. of VNI** | **8.1** | **23.8** |
| Tu_406_1 | VNII | 10.0 [4.9–18.5] | 0 |
| HamdanC3'1 | VNII | 9.3 [4.5–22.1] | 0 |
| | **Av. of VNII** | **9.6** | **0** |
| **All *C. neoformans* (Average)** | | **9.7** | **18.8** |
| (average) | | | |
| B3501 | VNIV | 7.2 [4.5–17.43] | 0 |
| CBS6995 | VNIV | 4.5 [3.4–11.1] | 0 |
| | **Av. of VNIV** | **5.85** | **0** |
| **All *C. deneoformans* (Average)** | | **5.85** | **14.2** |

## Titanisation in *C. gattii* is a polygenic trait

Several genetic regulators have been implicated in the control of titanisation in *C. neoformans* [6,7,13,15,24]. The interaction between these (nuclear genome-encoded) genetic regulators and mitochondrial activity has been proposed [16]. Consequently, we exploited a collection of parent/progeny crosses that we generated as part of an earlier study, and for which mitochondrial genotype is known [33], to begin to investigate the genetic control of titanisation in *C. gattii*. Mitochondrial genotype and inheritance were confirmed by the expression of *ATP6* gene (encoded by the mitochondrial genome) [33].

Firstly, we investigated a cross between *C. gattii* R265 and *C. gattii* LA584, a strain that belongs to the same VGII group as R265 but which shows a significantly lower capacity to titanise in our *in vitro* conditions (Fig 8A). The progeny from this cross showed considerable variation in titanisation capacity, with only one (Alg30) exceeding the titanisation capacity of R265. Ploidy measurement of parents/progeny based on flow cytometry analysis of DAPI staining for DNA content is shown in S6A Fig. Notably, there was no correlation between titanisation capacity of individual progeny and the mitochondrial genotype (R265 or LA584) that they had inherited, suggesting that mitochondrial genotype is not a major driver of this phenotype.

We then turned our attention to a cross between R265 and a more distantly related strain, B4564 (VGIII), which shows relatively low levels of titanisation. In this outgroup cross, a 100% inheritance of the mitochondrial genome from B4564 (*MAT***a**) was confirmed in 18 progeny

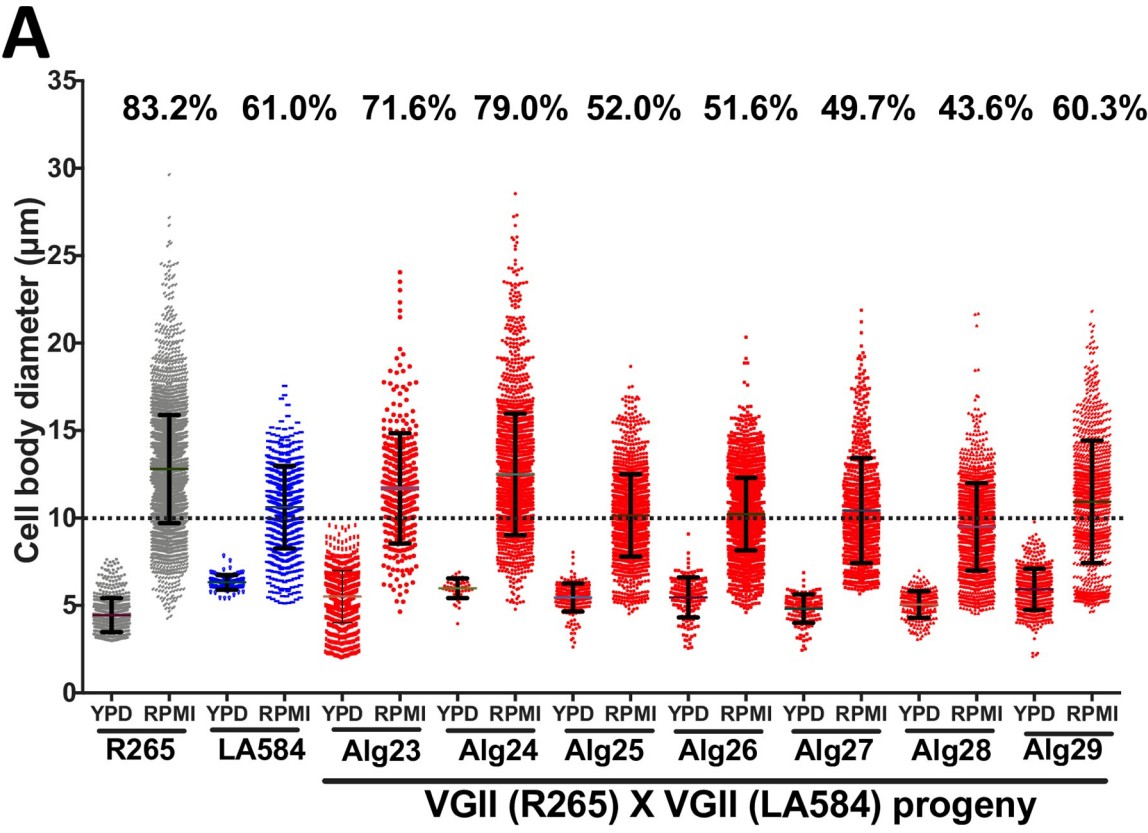

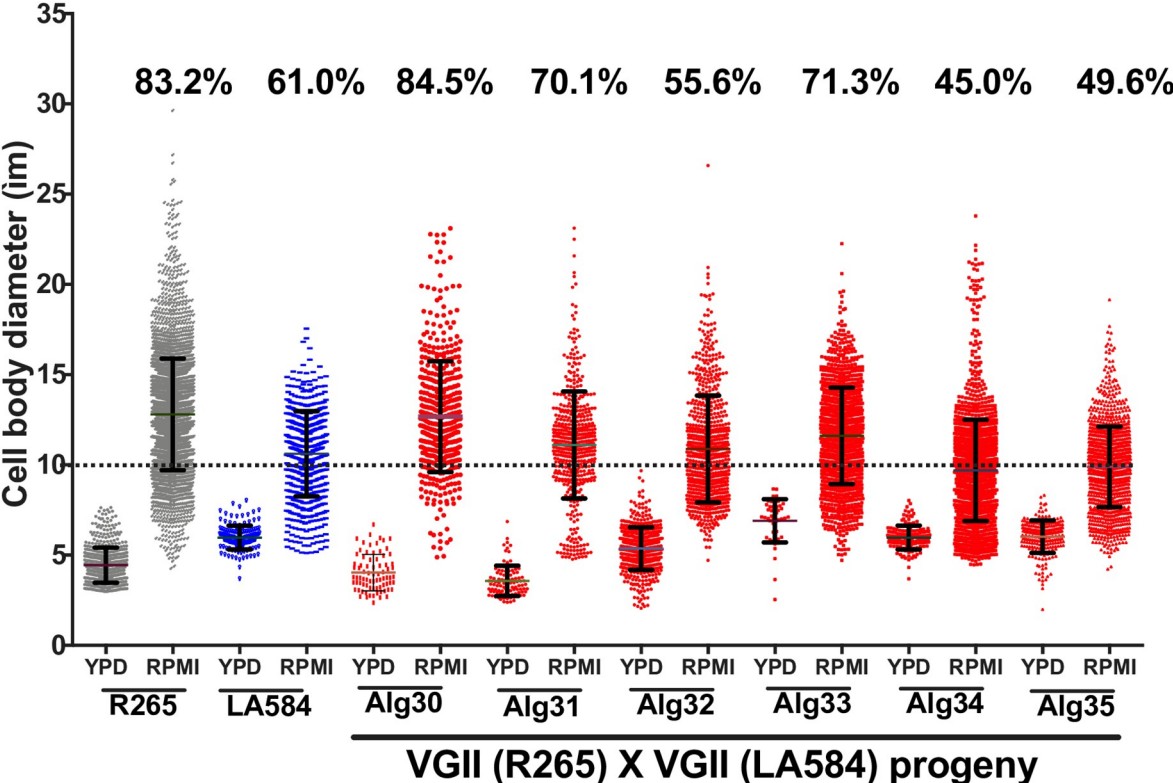

**Fig 8. Capacity to form titan cells of *C. gattii* progeny arising from two crosses.** (A) Titanisation pattern following three days of induction for R265 (VGII) x LA584 (VGII) and 13 progeny (Alg23-Alg35) arising from this cross [33]. (B) Titanisation pattern following three days of induction of R265 (VGII) x B4564 (VGIII) and 18 of the progeny (P1-P18) arising from this cross.

[33]. Despite this uniparental mitochondrial inheritance, all the progeny exhibit a significantly higher capacity for titanization than B4564 by both size and ploidy (Figs 8B and S6B). Furthermore, 12/18 progeny showed a higher capacity for titanisation than either parent (p<0.0001) with 5 progeny (P1, P5, P7, P8, P9 and P18) showing a remarkable >95% titan cell population by the end of the assay. Together, these data suggest that the nuclear genome, and not the mitochondrial genome, is the major source of variation in the capacity of different isolates to form titan cells.

## Discussion

*Cryptococcus* adaptation to the host environment is accompanied by phenotypic, metabolic and genetic alterations that are essential for pathogenicity [13,34–38]. Typically measuring 5–7 μm, the cryptococcal yeast cell can undergo a morphological switch in the lungs to form enlarged polyploid titan cells, which have been studied *in vivo* [6,7] and recently induced *in vitro* [13–15].

Here we report a new *in vitro* model for induction of *bone fide* titan cells. Our protocol is a one-step incubation of cryptococcal cells in serum-free RPMI media with 5% $CO_2$ at 37˚C and is therefore highly amenable to high-throughput screens.

Using our *in vitro* protocol, we conducted a detailed analysis of titanisation in the *C. gattii* strain R265 (VGIIa). We showed that R265 titan cells induced *in vitro* possess enlarged cell body size, a large central vacuole, thick capsule and cell wall, and were polyploid after 5 days. Consequently, they exhibit all the hallmarks of true titans produced during mammalian infection. Unlike other cryptococcal strains, however, R265 shows a separation between size and ploidy increase. We suggest that the asynchronous progression of these two events may be due to R265 undergoing cell size increase without passage through the cell cycle and then subsequently switching to DNA replication once the critical volume has been attained.

The lack of synchronisation between cell enlargement and DNA replication is a major difference between R265 titan cells and other well-studied *Cryptococcus* titan cells [6,7,13,14,16,24,27]. We suggest that the asynchronous progression of these two events is due to the prolonged time the titan cells spend in the G1 phase and/or cell cycle arrest at the G1/S checkpoint. This is supported by the ploidy of the 24 hr induction samples, where the vast majority of cells show a 1C DNA content. In yeast, a late G1 cell cycle arrest, known as "Start", is part of the core cellular response to stress [39,40]. In R265 we observe that cell enlargement occurs before polyploidization during the early period of induction, so that the enlarged cells remain as 1C haploid yeast cells for the first two days of induction. By day 3 of induction, the enlarged cells begin to duplicate their DNA content to at least 2C but almost completely stop budding at the same time [budding index: 2.8% (26/933)] (Fig 4C). This leads to a major distinction between the species, in that *C. neoformans* titan cells can produce daughter cells within our *in vitro* titan induction model (S7 Fig) whilst C. *gattii* R265 titans do not. We note that this observation may be of value in studying cell cycle dynamics in *Cryptococcus* species (particularly *C. gattii*), since producing synchronised cryptococcal populations for such investigations has previously been methodologically challenging.

To explain the unbudded phase of the induced cells, we attempt to correlate this phenotype with cell cycle progression with reference to *C. neoformans*. In *C. neoformans*, large unbudded G2 cells have been shown to emerge during a stationary growth phase [41]. Recently, while

scrutinizing the cell cycle regulation of titan cells in *C. neoformans*, Altamirano *et al.* [19] described a two-step process of titanisation: a) typically-sized cells duplicate DNA to 2C and arrest in G2 as unbudded cells; and b) then the cells are released (by the combined influence of the cell cycle gene Cyclin Cln1 and "stress signals") to form polyploid titan cells. Contrary to this phenomenon, R265 cells exhibit an actively budding phase in the early period of induction where the majority of the cells display 1C DNA content consistent with G1 and then undergo DNA replication to form G2 arrested unbudded polyploid titan cells at the later time point (day 5 onwards) where cells with 1C DNA content are absent (Fig 4B). In agreement with this observation, we observed that the transcriptional profile of cells undergoing titanisation mirrors their cell-cycle phenotype. Consequently, the budding and mitosis genes, *CDC11* and *CLN1*, were expressed early and peaked at 24hrs (a point at which budding is prevalent and most cells have a 1C DNA content), suggesting that the cells were predominantly in either G1 or M phase. *CLN1* is required for releasing *C. neoformans* titan cells from G2 arrest (during cell cycle progression) [18] and therefore it is not surprising for that *CLN1* is downregulated at day 3 (Fig 6B) when the budding index drops significantly (Fig 4C). However, it is intriguing that *CLN1* is partially upregulated at day 5 and 7. In *C. neoformans*, *CLN1* forms a critical balance between DNA replication and cell division [18]. Our data suggest that *CLN1* is involved in the regulation of cell division during the first 24 hr of induction and in DNA replication during the unbudded phases at 5 and 7 days.

Although we did not study cell cycle regulation of *C. neoformans* titan cells, *C. neoformans* titans generated via our *in vitro* system differ from R265 titan cell by profusely budding after 3 days of induction (S7 Fig). *C. neoformans* titan cells can pass through the G2/M checkpoint (G2/M transition), commit to mitosis and produce buds [6,7,13,15]. Interestingly, the G2 arrested R265 titan cells actively proliferate when recultured in nutrient rich media, suggesting that they are not permanently growth arrested but rather in quiescence. In budding yeast, quiescent cells are known to be G1 arrested during stationary phase [42] whereas both G1 or G2 arrest is possible for *Cryptococcus* [41]. Therefore, it is not surprising that R265 quiescent titan cells show a G2 arrest. However, the seemingly high metabolic activity of these cells (when compared to log-phase R265 yeast cells) is intriguing and warrants further investigation in the future.

We also demonstrate that R265 titan cells produce daughter cells with a polyploid DNA content similar to their mother cells. In contrast, *C. neoformans* titan cells produce both haploid and aneuploid progeny [24], with sizes ranging between 5–7μm and 2–4μm respectively. Given that population heterogeneity in *C. neoformans* is associated with preferential dissemination to the CNS [43,44], it is possible that this lack of heterogeneity in *C. gattii* may contribute to the differences in disease etiology in this species. In particular, murine models have shown that *C. neoformans* isolates that produce high percentage of titan cells fail to disseminate to the brain and instead remain in the lung, consistent with pneumonia or chronic infection [13,24].

Through our *in vitro* model, we confirmed the requirement of host-relevant environmental cues (physiological temperature of 37°C and 5% $CO_2$) for titan cell formation, as previously discovered [14,15]. We also highlighted cell density as a key regulatory factor for titan induction in *C. gattii*, as it is in *C. neoformans*. Consistent with previously reported *in vitro* induction model [14,15], production of R265 titan cells was inversely proportional to initial inoculum density such that no titan cells were produced at high density ($10^6$ cells/mL) and the maximum percentage was achievable at low density of $5 \times 10^3$ cells/mL. Interestingly, addition of the quorum-sensing peptide, Qsp1 [44,45] was able to significantly inhibit titan cell formation in our assay.

The fact that RPMI, but not the very similar cell culture medium DMEM, induced R265 titan cells enabled us to identify p-Aminobenzoic acid (pABA) as a major driver of titanisation. Interestingly, the titan induction effect of pABA was recapitulated when the *C. neoformans* strain, H99 was tested, although the very low titan formation capacity of this strain means that the effect was very weak (S1E Fig). The mechanism by which pABA triggers titanisation remains unclear at present. However, we note that pABA is an antifungal metabolite that has efficacy against several fungal plant pathogens such *Fusarium graminearum*, *Magnaporthe oryzae*, *Rhizoctonia solani*, *Sclerotinia sclerotiorum* and *Valsa ambiens var. pyri*. [46]. Since titanisation is linked to the fungal stress response, it may be that low dose pABA induces a mild stress that triggers titanisation. In this context modulation of the cell cycle and morphogenesis of *Colletotrichum fructicola* (a plant fungal pathogen) by pABA has been documented [47]. Alternatively, pABA's well documented role in oxidative damage tolerance and the role of reactive oxygen species in titan cell induction [22,47,28] may suggest a role for reactive oxygen balance in this phenomenon.

Finally, using our *in vitro* protocol, we evaluated the capacity for titan cell formation between and within cryptococcal species. We found titanisation was particularly abundant within *C. gattii*/ VGII (*C. deuterogattii*). Interestingly, Fernandes and colleagues reported a similar cell enlargement phenotype while screening for clinically relevant attributes in *C. gattii* [20]. To start to dissect the genetic regulation of this process, we tested and analysed the titanisation profile of a collection of parent/progeny crosses that we generated in our previous study [33]. It was striking to note the variation in this phenotype within recombinant progeny and, in particular, the very high rates of titanisation found in the offspring of 'outgroup' hybrids. Most of the progeny from this cross showed titanisation rates equal to or greater than that of R265 (the high titan cells generating parent). It is possible that this reflects a 'hybrid vigour' effect, resulting from the outcross. In the future, more detailed genomic investigation of these and other crosses may potentially facilitate a more comprehensive understanding of titan cell formation in this genus.

Overall, our titan induction protocol is an efficient and high throughput approach for producing titan cells at scale. By employing our *in vitro* protocol, we have discovered novel aspects of titanisation in *C. gattii* and revealed the separation of DNA replication and cell size increase. Together, we hope that this approach will provide a platform for the future mechanistic investigation of titanisation in this important group of pathogens.

## Methods

### Cryptococcal strains and culture conditions

Cryptococcal strains used in this study are listed in Table 2. Prior to use, cryptococcal strains were maintained on Yeast Peptone Dextrose (YPD) (1% yeast extract, 2% bacto-peptone, 2% glucose, 2% bactor-agar) agar at 4°C from which overnight cultures were prepared in YPD broth at 25°C, 200rpm.

### *In vitro* induction of Titan cells

Yeast cells from overnight cultures were collected (by centrifugation at 4000 r.p.m for 2 mins), washed three times with phosphate buffered saline (PBS), re-suspended in 3mL PBS and counted on a haemocytometer to determine cell densities. Except where otherwise noted, titan induction was achieved by inoculating $5 \times 10^3$ yeast cells in 1 mL serum free-RPMI 1640 within a 24 well tissue culture plate for 24 hours to 21 days at 37°C in 5% CO2 without shaking.

For generation of daughter cells, the 7 day old titan-inducing culture was passed through a >20μm cell strainer and >20μm-sized Titan cells were re-cultured in YPD overnight. Then, the daughter cells were isolated by filtering the overnight culture using a 15μm cell strainer and collecting the flow-through.

**Table 2. Cryptococcal species and strains used in this study.**

| Species and strains | Serotype | Genotype | |
|---|---|---|---|
| *C. gattii* WM265 | B | VGI | Clinical isolate, Brazil |
| *C. gattii* WM179 | B | VGI | Clinical isolate, Australia |
| CBS8755 | | VGI | |
| C384 | | VGI | |
| NIH312 | C | VGI | Clinical |
| B4546 | C | VGI | Clinical |
| EJB11 | B | VGI | |
| MMCO8-897 | B | VGI | |
| CBS1508 | B | VGI | |
| CBS1873 | B | VGI | |
| *C. gattii* R265 | B | VGIIa | Clinical isolate, Vancouver, Canada |
| *C. gattii* CDCR271 | B | VGIIa | Clinical isolate, immunocompetent patient, Kelowna, British Columbia, Canada |
| *C. gattii* ENV152 | B | VGIIa | Environmental isolate, Alder tree, Vancouver Island, Canada |
| *C. gattii* ICB180 | B | VGII | Environmental isolate, Eucalyptus tree, Brazil |
| *C. gattii* CBS10089 | B | VGII | Clinical isolate, Brazil |
| *C. gattii* CDC272 | B | VGII | Clinical isolate, Greece |
| *C. gattii* B7735 | B | VGIIb | Clinical isolate, Vancouver, Canada |
| *C. gattii* EJB11 | B | VGIIb | |
| *C. gattii* EJB52 | B | VGIIc | Clinical isolate, Oregon, USA |
| *C. gattii* CBS6955 | B | VGIII | Clinical isolate, Oregon, USA |
| CBS6993 | C | VGIII | Clinical isolate, USA |
| CBS1622 | B | VGIII | |
| WM1243 | B | VGIII | |
| B13C | | VGIII | Clinical isolate, Asia |
| CA2350 | | VGIII | Clinical isolate |
| CA1227 | | VGIII | Clinical isolate |
| *C. gattii* LA584 | B | VGII | Clinical isolate, Colombia |
| *C. gattii* B5464 | C | VGIII | Clinical isolate, USA |
| *C. gattii* WM779 | C | VGIV | Clinical isolate, USA |
| B5742 | B | VGIV | |
| B5748 | B | VGIV | |
| CBS10101 | C | VGIV | |
| *C. gattii* CBS1010 | C | VGIV | Veterinary, South Africa |
| *C. neoformans* H99 | B | VNI | Clinical isolate, USA |
| *C. neoformans* Zc1 | A | VNI | |
| *C. neoformans* Zc8 | A | VNI | Clinical, Zambia |
| *C. neoformans* Z12 | A | VNI | Clinical, Zambia |
| *C. neoformans* 125.91 | A | VNI | Clinical, Tanzania |
| *C. neoformans* Tu369-2 | A | VNI | Environmental isolate, Mopane tree bark, Botswana |
| *C. neoformans* HAMDANC 3–1 | A | VNII | Pigeon droppings, Belo Horizonte, Brazil |
| *C. neoformans* B3501 | D | VNIV | Clinical isolate, USA |
| *C. neoformans* CBS6995 | D | VNIV | Clinical isolate, USA |
| *C. neoformans* CBS8336 | A | VNI | Wood of Cassia tree, Brazil |

## Cell size measurement

Cells recovered from titan induced or YPD grown cultures were washed in PBS and fixed with 50% methanol. After Indian ink staining for capsule visualization, cellular images were

obtained using a Nikon TiE microscope equipped with phase-contrast 20X optics. Cell body and capsule sizes for individual cells were measured by using ImageJ software in combination with automated measurement based on Circle Hough Transformation algorithm [49–51].

## Cell wall and capsule

Cells were fixed with 4% methanol-free paraformaldehyde for 10 mins and stained with calco-fluor white (CFW, 10 μg/ml) for another 10 mins [13]. Total chitin was determined by flow cytometry on an Attune NXT instrument, with quantification of CFW staining using FlowJo software. For capsule visualization, cells were counterstained with India Ink (Remel; RMLR21518) and images acquired using a Nikon TE2000 microscope and analysed using ImageJ software.

## Quorum sensing effect

The quorum sensing Qsp1 (NFGAPGGAYPWG) and scrambled Qsp1 [(AWAGYFPGPNG), control] peptides were synthesized at the University of Birmingham and then dissolved in ster-iled distilled water at 1mM and frozen at -20˚C for future use. The peptides were then added to RPMI media at concentration of 20μM and R265 yeast cells from overnight culture were grown in the mixture at 37˚C in 5% $CO_2$ for 72 hr. Finally, the cell body size of R265 cells in the test media were analysed.

## DNA content measurement Ploidy

RPMI and YPD grown cells were recovered, washed 3x in PBS, fixed in 50% methanol and stained with 3 ug/ml DAPI at $10^5$ cells/mL. For each sample, about 10000 cells were acquired on an Attune NXT flow cytometer and the result was analysed using FlowJo v. 10.7.1. Cells were sorted for doublet and clump exclusion by using FSC-A vs FSC-H gating strategy (S3E Fig) and compared to control, YPD grown, yeast cells.

## Identification of RPMI-titan inducing compound(s)/factors

RPMI-specific compounds (which are absent in DMEM) were supplemented to DMEM media and tested for capacity to induce titan cells in R265 (*C. gattii*). Accordingly, DMEM media was first supplemented with '**RPMI-specific' amino acids** at their original concentration in RPMI: L-Glutamic acid (20 mg/L), L-Aspartic acid (20 mg/L), L-Arginine (200 mg/L), L-Glutathione (1 mg/L), L-Asparagine (50 mg/L) and L-Proline (20 mg/L). All the amino acids were purchase in powder form from Sigma-Aldrich and stock solutions were made by dissolving them according to manufacturer's instruction.

To investigate if the **glucose concentration difference** between RPMI (high concentration) and DMEM (low concentration) was responsible for titan cell formation, DMEM was supple-mented with D glucose (Purchase from Sigma-Aldrich) at a final concentration of 2000 mg/L (RPMI concentration). D glucose was purchased in powder form and stock solution was pre-pare according to manufacturer's instructions.

Finally, DMEM media was first supplemented with **other 'RPMI-specific' compounds** at their original concentration in RPMI: Vitamin B12 (0.005 mg/L), Biotin (0.2 mg/L), and para-aminobenzoic acid [(pABA) 1.0 mg/L]. All these compounds were purchased from Sigma-Alrich in powder form and stock solutions were prepared according to manufacturers instructions.

## Metabolic activity of titan

The metabolic state of R265 titan cells was confirmed by using FUN-1 reporter dye. The freshly obtained RPMI-induced 7 day old titan and YPD overnight (control) cells cultures were collected and aliquoted into two. Cells from an aliquot from the two conditions were heat-killed for 1 hr in a heating block (Stuart Heating Block). Both heat-inactivated and fresh cell samples were stained with Fun-l at 5μM for 30 mins. Microscopy images were taken with Zeiss Axio Observer at 63X magnification with fluorescent emission wavelength set at 590 nm.

## RNA extraction and purification

Total RNA extraction was performed on uninduced (YPD grown) and titan-induced R265 cells by employing the protocol of QIAGEN (RNeasy Micro Kit [45]); Cat. No./ID74004) with slight modification. Samples of overnight YPD grown, R265 cultures and titan-induced cells of the different time-points (24 hr, 72 hr, 5 days and 7 days) were harvested, washed three times in PBS, adjusted to ~$10^6$ cells/mL and pelleted in 1.5mL Eppendorf tubes. The cell pellets were flash-frozen in liquid nitrogen and stored at -80˚C overnight. The cells were lysed by mixing in 400μL of RLT buffer, transferring to a 2 mL lysing tubes (MP Biomedicals 116960100) and beating with a bead beater (MP Biomedicals 116004500 FastPrep 24 Instrument Homogenizer) for thorough cell disruption. The homogenized sample was centrifuged for 3 min at 10,000 xg at room temperature. The aqueous portion was separated, mixed with 70% ethanol at 1:1 ratio and transferred to an RNeasy Mini Spin Column. Finally, RNA extraction and purification were carried out as described in manufacturer's protocol, including purification through a gDNA eliminator column to remove DNA contamination.

## RT-qPCR and gene expression analysis

RNA was extracted from YPD grown and titan-induced samples and reversed transcribed (RT) to cDNA by using FastSCRIPT cDNA Synthesis protocol (Catalogue Number: 31-5300-0025R]. In brief, 15μL of RNA samples were mixed with 1μL of RTase and 4μL of FastSCRIPT cDNA Synthesis Mix (5X) before 30 min incubation at 42˚C and subsequent 10 min incubation at 85˚C. Quantitative PCR for the selected putative cell cycle genes was determined for each RT samples by mixing 2μL of the RT samples with 38μL master mixed of KAPA enzyme (KABA SYBR FAST qPCR Kits) and designed primers (S1 Table) and run in a real-time PCR detection system (CFX96 Touch Real-Time PCR Detection System; Ref. no.: 1845096). Gene expression level was obtained and normalized according to change difference with the housekeeping gene, *GAPDH*. Finally, the relative expression profile was expressed as a function of comparative threshold cycle ($C_T$) by using the follow formula (III):

i.  Delta $C_T = C_{T\,gene} — C_{T\,GAPDH}$

ii.  Delta delta $C_{T\,=}\,C_{Tgene\,-}$average Delta $C_T$

iii.  Relative gene expression = $2^{-delta\,deltaCt}$

## Statistical analysis

All analyses were performed with GraphPad Prism 6. The Shapiro-Wilk test was used to check normality of the data and either parametric (ANOVA) or non-parametric (Kruskal-Wallis or Mann Whitney, as described) tests used accordingly. Size distribution plots are shown as scatter plots with the mean highlighted and error bars reflecting the standard deviation.

## Supporting information

**S1 Fig. Phenotypic analysis and impact of exogenous factors on titan cell induction in *C. gattii*. A)** Cell body size of R265 yeast cells before (in YPD) and after incubation in sterile RPMI and serum supplemented RPMI. The cells were grown in YPD overnight and in RPMI (sterile and serum-contained) for 3 days in 5% $CO_2$ at 37˚C and recovered for cell body size measurement. **B)** Micrograph of R265 yeast cells after grown in YPD for 7 days in 5% $CO_2$ at 37˚C. Scale bar = 15μm. **C)** pH of YPD amd RPMI before and during titan induction (24 hr and 7 days). Titan cell induction was performed with the R265 (*C. gattii*) and pH was measured. **D)** Effect of glucose on titan cell formation. DMEM media was supplemented with D glucose reaching 2000mg/L (the concentration in RPMI) and tested for capacity to induce cell enlargement (>15μm) as compared to RPMI. Titan induction was performed by incubating R265 yeast cells in the different media conditions for 24 hrs at 37˚C in 5% $CO_2$. **E)** The influence of pABA on titan induction in *C. neoformans* (H99). Cell body size was measured after H99 yeast cells were grown in the different induction media at 37˚C in 5% $CO_2$ for 72 hr. (DOCX)

**S2 Fig. Effect of $CO_2$ and reculturing on titan phenotypes. A)** Effect of 5% $CO_2$ on cryptococcal growth. Cells incubated at 37˚C in serum-free RPMI in 5% $CO_2$ or atmospheric conditions for 24 hrs were assessed for colony forming unit (CFU) on YPD after to evaluate viability. Statistical significance was confirmed by a Two-tailed t-test where ** = p<0.05. **B)** Proliferation of R265 Titan cells after re-culturing in RPMI. Titan cells obtained from induced cultures after 7 days (A) were re-cultured in fresh, serum-free RPMI at 37˚C in 5% $CO_2$ for 24 hrs (B) and analysed microscopically for ability to bud. Scale bar = 15μm. (DOCX)

**S3 Fig. Characterization of budding nature and titanisation of R265 titan-derived daughter cells. A)** Images showing the budding nature of daughter cells before and after titan induction for 24hr, 72 hr and 7 days. Scale bar = 5μm. **B)** Budding index of daughter cells before and after titanisation. **C)** Cell body diameter was measured microscopically, and percentage of titan cells determined based on >10μm cell size. The data represents three independent biological repeats and significance was confirmed by one-way ANOVA where, **** = p<0.0001 **D)** Daughter cells of R265 titan cells were isolated and returned to titan inducing condition and their ploidy was determined by DNA content measurement via flow cytometry before titan induction (green), after 24 hr (blue) and 7 days (brown) relative to R265 haploid yeast (red). **E)** Flow cytometry data showing the gating strategy employed to confirm cell ploidy of R265 titan daughter cells as compared to YPD grown yeast and (>20μm) filtered titan cells. (DOCX)

**S4 Fig. Cell body diameter of 42 YPD grown and titan-induced cryptococcal isolates representing the different genotypes within the *C. neoformans/gattii* species complex.** All strains were induced for titan cell formation according to our *in vitro* protocol. The cell body diameter and ploidy (see S5 Fig) were determined after 72hrs as described in the Methods section. A, B, C, D) Cell body size distribution of isolates from *C. gattii* complex (VGI-VGIV) before induction (YPD) and after 72 hr of induction (RPMI). E, F) Cell body size distribution of isolates from *C. neoformans* (VNI-VNII) and *C. deneoformans* (VNIV) before (YPD) and after 72 hr induction (RPMI) respectively. (DOCX)

**S5 Fig. DNA content of 42 YPD grown (red) and titan-induced (blue) cryptococcal isolates representing the different genotypes within the *C. neoformans/gattii* complex.** All isolates were induced for titanisation according our *in vitro* induction model (as mentioned in the Methods

sections) and DNA content was assessed by DAPI staining and flow cytometry analysis.
(DOCX)

**S6 Fig. Ploidy of parent and progeny *C. gattii* strains before and after titan induction. A)** DNA content of YPD grown (red) and titan-induced (blue) of ingroup crossing [VGII(R265) x VGII (LA584)] strains and their 13 progeny (Alg23-Alg35) after 3 days. All isolates were induced for titanisation according to our *in vitro* induction model (as mentioned in the Methods sections) after DNA content was confirmed by DAPI staining and flow cytometry analysis. **B)** DNA content of YPD grown (red) and titan-induced (blue) of ingroup crossing [VGII (R265) x VGIII (B4564) strains and their 18 progeny (P1-P18) after 3 days. All isolates were induced for titanisation according to our *in vitro* induction model (as mentioned in the Methods sections) after DNA content was confirmed by DAPI staining and flow cytometry analysis.
(DOCX)

**S7 Fig. Morphology of *C. neoformans* cells before and after titan-induction via our *in vitro* model.** Microscopy images of YPD grown **(A)** and 72 hr titan-induced *C. neoformans* cells **(B).** Scale bar = 5μm.
(DOCX)

**S1 Table. List of cell cycle phenotypes of titan induced and associated genes.**
(DOCX)

**S1 Movie. Example timelapse movie showing titan cell formation in *C. gattii* R265 under *in vitro* conditions.**
(AVI)

## Author Contributions

**Conceptualization:** Leanne Taylor-Smith, Elizabeth R. Ballou, Robin C. May.

**Formal analysis:** Maria Makarova, Elizabeth R. Ballou.

**Funding acquisition:** Robin C. May.

**Investigation:** Lamin Saidykhan, Joao Correia, Andrey Romanyuk, Guillaume E. Desanti, Maria Makarova.

**Methodology:** Lamin Saidykhan, Joao Correia, Anna F. A. Peacock, Maria Makarova, Robin C. May.

**Project administration:** Elizabeth R. Ballou, Robin C. May.

**Resources:** Anna F. A. Peacock, Robin C. May.

**Supervision:** Guillaume E. Desanti, Leanne Taylor-Smith, Maria Makarova, Elizabeth R. Ballou, Robin C. May.

**Validation:** Robin C. May.

**Writing – original draft:** Lamin Saidykhan.

**Writing – review & editing:** Anna F. A. Peacock, Elizabeth R. Ballou, Robin C. May.

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
