## [Decision Letter · Decision Letter 0]

24 Feb 2022

Dear Prof May,

Thank you very much for submitting your manuscript "An in vitro method for inducing titan cells reveals novel features of yeast-to-titan switching in the human fungal pathogen Cryptococcus gattii" for consideration at PLOS Pathogens. As with all papers reviewed by the journal, your manuscript was reviewed by members of the editorial board and by several independent reviewers. In light of the reviews (below this email), we would like to invite the resubmission of a significantly-revised version that takes into account the reviewers' comments.

While all of the reviewers felt that much of the data presented in the manuscript was sound, there were significant concerns raised regarding the novelty of the findings. Specifically, the production of titan cells by *C. gattii* has already been demonstrated and it was felt that the authors did not provide sufficient comparison to *C. neoformans* titan cell formation to show that the differences in induction of titan cells they observed with *C. gattii* would not also occur with *C. neoformans*. As one example, without assessing the role of p-aminobenzoic acid in *C. neoformans* titan cell production it is difficult to determine the novelty of this finding to *C. gatti*i. In addition, the data relating to the metabolic state of the cells, and whether the cells analyzed were in fact dead, raised additional questions about the novelty of these findings. While there were novel aspects of the study, such as the genetic crosses, it was felt that overall these were minimal in nature and thus did not raise the level of significance of the study to a level appropriate for publication in PLoS Pathogens. Some of the reviewers felt that the manuscript was interesting, and that appropriate inclusion of additional data would strengthen the significance and novelty of the observations to the broader PLoS Pathogens audience. It should be noted that this would require major revision of the manuscript and inclusion of additional experimental data.

We cannot make any decision about publication until we have seen the revised manuscript and your response to the reviewers' comments. Your revised manuscript is also likely to be sent to reviewers for further evaluation.

Sincerely,

Kirsten Nielsen, Ph.D

Guest Editor

PLOS Pathogens

Xiaorong Lin

Section Editor

PLOS Pathogens

Kasturi Haldar

Editor-in-Chief

PLOS Pathogens

orcid.org/0000-0001-5065-158X

Michael Malim

Editor-in-Chief

PLOS Pathogens

orcid.org/0000-0002-7699-2064

Reviewer's Responses to Questions

**Part I - Summary**

Reviewer #1: THe authors were interested in characterrizing Titan cell formation in C. gattii based on a simple standardizable protocol. They brought new data on the genetics of TC formation using progeny experiments.

Reviewer #2: This study demonstrates that in vitro induction of the crucial titan morphotype in Cryptococcus neoformans extends to C. gattii. While this conservation itself isn’t a major surprise, the authors then demonstrate that differences in when and how C. gattii enters the titan state compared to C. neoformans is very biologically interesting, including the decoupling of polyploid formation with cell growth and budding. In addition, the seemingly quiescent state of the C. gattii titans raise additional questions about whether these cells are metabolically active. Depending on the metabolic activity of these cells, titan formation in C. gattii could include a persister-like state that could have fascinating biological implications.

Reviewer #3: The present manuscript describes the formation of titan cells in the fungal Cryptococcus gattii. This process has already been extensively described in C. neoformans, in particular in 2018 when three different groups described different in vitro conditions to induce this process.

The main strenght of the article is the description of all the ploidy changes associated to titan cell formation in C. gattii, and in particular, the differences with C. neoformans.

While the present article describes a new method to induce this process, many of the factors described have already been characterized in C. neoformans (temperature, strain variation, CO2, cell density effect, among others). For this reason, the novelty of the manuscript is limited. It has been also shown that C. gattii, and in particular this strain (CBS10514, R265) can form titan cells in vitro (reference 14).

Most of the results, and in particular those related to the ploidy and cell cycle, have been performed using a C. gattii isolate (R265, CBS10514), which was originally isolated from the Vancouver outbreak. This strain has been characterized in detail, and it has been shown that it was the result of a very rare event of same sex mating in fungi (Nature volume 437, pages 1360–1364 (2005)). For this reason, I feel that it is too premature to extrapolate the mechanisms described to the whole Cryptococcus gattii species, and ploidy and cell cycle mechanims should be described in more strains from different genotypes.

In general, the experiments are well planned and perform, although I feel that new experiments to address the specific mechanism required for titanisation in C. gattii should be performed.

**Part II – Major Issues: Key Experiments Required for Acceptance**

Reviewer #1: Incubation in YPD at 37°C and 5% CO2 (24H and 7 days) as a control condition of titan cell formation would be a nice addition to the author’s experiments. Add to Fig 1, 3D, 6. Indeed YPD at 25° is very different conditions. It should be interesting to identify the most important factors associated with Titan cell formation in C. gattii (37 and 5% C02 vs. medium??)

Line 87-88: could you add details on the use of serum in the first induction medium as in cell culture medium, fetal calf serum is often added to help mammalian cells to grow.

Line 115: All in vitro protocols evidenced a large vacuole occupying the cytoplasmic space.

Line 129-130: Fig 1E: the 3 large cells are looking as dead cells. Indeed, the nice phenotype inclufing the large vacuole is not visible on this picture. These cells have been characterized in Dromer’s lab pmid: 25827423 and look similar.

Please use merge picture (DIC+DAPI) for DAPI stainings. This will better show that the DAPI staining does not stain a nucleus but rather the remaining retracted cytoplasm of this dead cell. This point is important, because to describe a population with 16C, the authors should be sure that the staining is done on a population harboring living cells in a good shape.

More pictures of 7 days incubated C. gattii using DIC should be provided. Fig 1A are only showing 24h-incubated cells

Fig 4E: The authors should be sure that no “contamination with titan cells” occurs in their setting to avoid detecting polyploid Titan cells within the population of daughter cells. Please analyze and provide histograms of all population (in SSC and FSC channels) to assess the absence of big cells from the daughter cells purification.

Line 238-241. It is not clear in the text which conditions have been used for RNASeq analysis. Between 24h and 7days or at 24h and 7d?

Figure 6 (impact cell density and temperature and CO2) should appear earlier in the results. As these conditions should be fixed rapidly to standardize experiments and obtained the optimal Titan cell proportion.

Line 323: should be 6F instead of 6B

Fig 6G and 6H should be separated from the rest of Fig 6.

Fig 6H. why using 3 h rather than 7 days as the best time point to obtain a large number of TC.

The authors should investigate the pH of the different media tested. pH has an impact on Titan cell formation. Fig 6H that authors should investigate why pABA acts as an inducer of TC. The authors could add pABA in YPD in addition as another control to validate the impact of pABA on C. gattii TC formation.

As the authors tend to show that VGII produce more TC than the other lineages, they should include more strains to validate this observation on more than 9 cells. They also should include at least the same number of strains from the other lineages (only 2 strains for VGI or VGIII, VGIV). Indeed, VGIII VGIV or VGI are producing TC in the same range than that of VGII strains. May be just a matter of number of strains characterized.

Reviewer #2: Overall, I think that this manuscript is scientifically sound and interesting. Some additional contrasts to C. neoformans (see below) would increase the broad appeal of this paper. This manuscript also raises the following points:

Major points:

Lines 89, 96: The use of the term “giant” cells here is unclear. Since one of the first two groups to identify titan cells referred to them as “giant” cells, this could be interpreted as another term for titans rather than large cells that are not yet conclusively titan.

The data presented in figure 2 demonstrating that titan cells increase capsule size but do not appear to be actively growing is fascinating. Are these cells metabolically active? Could they represent a fungal phenomenon analogous to bacterial persister cells? Do titans have a long chronological lifespan compared to cells grown long-term in RPMI?

Does pABA induce titanisation in C. neoformans? Does iron block titanisation in C. neoformans?

Reviewer #3: - The ploidy study should be confirmed in more strains, not only R265.

- The role of p-aminobenzoic acid should be further explored. They should validate some of the hypothesis from the discussion. Does p-aminobenzoic acid have any effect on other strains and species, such as C. neoformans?

- It is not clear how this protocol compares to those described in 2018. Also, I think that the authors should take advantage of this protocol to provide insights on the molecular mechanisms that differentiate titan cell formation in C. gattii compared to C. neoformans. To increase the significance of the article, I would strongly recommend to perform an RNAseq experiment and look at global changes in gene transcription.

- Line 84, results. The authors state that the discovery that C. gattii forms titan cells in RPMI was observed during experiments in which the yeasts were cocultivated with alveolar macrophages. Was serum present in the initial experiment? It is surprising that the authors do not evaluate the effect of serum in their experiments, since this was one of the main inducing factors that induce in vitro titanisation in C. neoformans in two of the protocols described in 2018.

**Part III – Minor Issues: Editorial and Data Presentation Modifications**

Reviewer #1: Line 30-31 : The critical role of titan cells for Cryptococcus neoformans to establish infection should be tempered.

Line 33. The discovery of this protocol is probably less serendipitous than the first observations in mice and for C. neoformans in vitro protocols.

Reviewer #2: Minor points:

The bars representing the median (? mean? not stated; should be) are very difficult to see in Figure 6H.

Reviewer #3: - Introduction should be rewritten. In particular, the authors should give more background, and summarize the findings of the article.

- The authors should revise the English style, and make sure that past tense is always used when explaning the results. Furthermore, they should avoid the use of "giant" cells, since it can be confused with "giant" macrophages.

- Statistics. The authors should include a specific statistics section in M&M. Furthermore, they have used ANOVA throughout the manuscript. However, this test can only be used when there is a normal gaussian distribution. And it seems that some of the size distribution are not normally distributed. They should assess the normality of all the samples (Shapiro-Wilk or Kolmogorov-Smirnov tests, and then evaluate whether a parametric (ANOVA) or non-parametric (i.e. Kruskal-Wallis) test should be applied.

- The definition of titan cell and thresholds used is confusing. In some cases, they use a threshold of 10 um, and in other figure they use a threshold of 15 um. This should be unified to 10 um and the data recalculated in all the figures (i.e, figure 6). If the threshold defined by the authors for titan cells is 10 um (cell body diameter), why do they use filters of 15 um pore size?

- It would be interesting to know if progeny from titan cells have a different chitin content compared to regular cells.

- I think that it is more correct to show the cell cycle and ploidy data (DAPI histograms) in lineal axis and not logarithmics, since the increse in DNA content is lineal.

- Figure 6. I would argue against the concept that C. gattii produces large haploid but non-titan cells. Titan cells should be defined by their size, since it is possible to obtain titan cells in different ways and through different mechanisms. I feel that from an imunological point of view, the main factor that determines immunological evasion is the size and not DNA content.

- Figure 6G. The authors should show the size of cells grown in YPD too.

- Regarding cell density, it has been shown in C. neoformans that cell density and quorum sensing phenotypes depend on a short peptide (QSP1), and that this peptide inhibits in vitro titanisation in C. neoformans (reference 14 and 15). I think that the authors should investigate if this peptide or other QS molecules have a similar effect in TC formation in C. gattii.

- Presentation of figure 5 is somehow confusing. I believe that it would be better to normalize the expression level in YPD to an arbitrary value of 1, and then present the rest of the data as fold-change compared to this condition.

- Table 2. Cryptococcus neoformans H99 strain is serotype A, VNI. This should be corrected.

- Line 576. The authors state in the RNA isolation section that it was done in 400 uL of "RNAase". Do they mean "RNAeay buffer"?

- RT-qPCR experiments. It seems that the samples were not treated with DNAse to ensure that there is not interference in the RT_qPCR of contaminant DNA. If this step was omitted, they authors should have carried out a control sample in which the RNAs are equally treated, but without the addition of reverse transcriptase, to make sure that these samples are negative. I recommend to confirm this, since this step can be very tricky, and DNA contamination can occur.

- Figure 4. It would be interesting to know what happens when titan cells recovered by filtration are incubated in fresh YPD and RPMI medium. Most probably, these cells are nutrient-depleted by day 2 of incubation, so it raises the issue of how do they behave when they are incubated again in a inducing and non-inducing medium.

- RPMI vs DMEM experiments. The authors should give the exact reference of the media used, and in particular, which glucose concentration do they contain. Furthermore, the authors should ensure that during the whole experiments (even through day 7), there is not a change in pH, and that these two media have the same buffering strenght for C. gattii. Yeast can significantly alter the pH medium by the secretion of protons or ammonia, and I am not sure how these media can buffer in these conditions without the addition of an additional buffer.

- Lines 358-363. The authors demonstrate that addition of iron inhibit titan cell development in C. gattii. This was also shown in C. neoformans (reference 14), so this should be highlighted.

PLOS authors have the option to publish the peer review history of their article (what does this mean?). If published, this will include your full peer review and any attached files.

Reviewer #1: **Yes: **Alexandre Alanio

Reviewer #2: No

Reviewer #3: No
---

## [Decision Letter · Decision Letter 1]

7 Jul 2022

Dear Prof May,

We are pleased to inform you that your manuscript 'An in vitro method for inducing titan cells reveals novel features of yeast-to-titan switching in the human fungal pathogen Cryptococcus gattii' has been provisionally accepted for publication in PLOS Pathogens.

Best regards,

Kirsten Nielsen, Ph.D

Guest Editor

PLOS Pathogens

Xiaorong Lin

Section Editor

PLOS Pathogens

Kasturi Haldar

Editor-in-Chief

PLOS Pathogens

orcid.org/0000-0001-5065-158X

Michael Malim

Editor-in-Chief

PLOS Pathogens

orcid.org/0000-0002-7699-2064

Reviewer Comments (if any, and for reference):

Reviewer's Responses to Questions

**Part I - Summary**

Reviewer #1: (No Response)

Reviewer #2: I think that the authors did an excellent job of addressing reviewer comments. The new data, particularly the revised figure 5, satisfied my concerns and make a good case for their scientific arguments.

Reviewer #3: The authors describe in this work in vitro conditions that efficiently induce the formation of titan-like cells in C. gattii. In the revised version, the authors have addressed most of the initial comments. I still believe that the main weakness is the lack of novelty because many of the factors described have already been identify in the literature as regulators of titan cells in C. neoformans. But even with this limitation, the authors provide a huge amount of information about titanization, and I believe that the article is significantly improved

**Part II – Major Issues: Key Experiments Required for Acceptance**

Reviewer #1: (No Response)

Reviewer #2: none

Reviewer #3: Most of them have been addressed in the revision

**Part III – Minor Issues: Editorial and Data Presentation Modifications**

Reviewer #1: (No Response)

Reviewer #2: none

Reviewer #3: Nothing to note

PLOS authors have the option to publish the peer review history of their article (what does this mean?). If published, this will include your full peer review and any attached files.

Reviewer #1: **Yes: **Alexandre Alanio

Reviewer #2: **Yes: **Jessica Brown

Reviewer #3: No

---

## [Editor Report · Acceptance letter]

12 Aug 2022

Dear Prof May,

We are delighted to inform you that your manuscript, "An in vitro method for inducing titan cells reveals novel features of yeast-to-titan switching in the human fungal pathogen Cryptococcus gattii," has been formally accepted for publication in PLOS Pathogens.

Best regards,

Kasturi Haldar

Editor-in-Chief

PLOS Pathogens

orcid.org/0000-0001-5065-158X

Michael Malim

Editor-in-Chief

PLOS Pathogens

orcid.org/0000-0002-7699-2064